# pH-Responsive Polypeptide-Based Smart Nano-Carriers for Theranostic Applications

**DOI:** 10.3390/molecules24162961

**Published:** 2019-08-15

**Authors:** Rimesh Augustine, Nagendra Kalva, Ho An Kim, Yu Zhang, Il Kim

**Affiliations:** BK 21 PLUS Center for Advanced Chemical Technology, Department of Polymer Science and Engineering, Pusan National University, Geumjeong-gu, Busan 46241, Korea

**Keywords:** controlled release, nanocarriers, polypeptides, responsive polymers, smart materials, targeted drug delivery

## Abstract

Smart nano-carriers have attained great significance in the biomedical field due to their versatile and interesting designs with different functionalities. The initial stages of the development of nanocarriers mainly focused on the guest loading efficiency, biocompatibility of the host and the circulation time. Later the requirements of less side effects with more efficacy arose by attributing targetability and stimuli-responsive characteristics to nano-carriers along with their bio- compatibility. Researchers are utilizing many stimuli-responsive polymers for the better release of the guest molecules at the targeted sites. Among these, pH-triggered release achieves increasing importance because of the pH variation in different organ and cancer cells of acidic pH. This specific feature is utilized to release the guest molecules more precisely in the targeted site by designing polymers having specific functionality with the pH dependent morphology change characteristics. In this review, we mainly concert on the pH-responsive polypeptides and some interesting nano-carrier designs for the effective theranostic applications. Also, emphasis is made on pharmaceutical application of the different nano-carriers with respect to the organ, tissue and cellular level pH environment.

## 1. Introduction

Currently, the number of people who are suffering with various diseases is increasing due to the unhealthy lifestyle, environmental pollution and adulteration of food. Thus, treatment of these various diseases requires the application of many drugs, gene delivery, and many other techniques for the accurate checking of the treatment progress. In most cases the treatment will affect the patient’s body as side effects of the drug or treatment itself and most of the drugs are not soluble in water and hence cannot be used directly [1,2,3,4,5]. The non-targetability of the drugs affects the normal cells or organs instead of the diseased cells or organs. Thus, the development of suitable carriers that can assure the bioavailability by maintaining the drug concentration in blood at the required level with a prolonged circulation time and can release the drug at a specific site by undergoing some morphological changes with respect to the physiological environment in the diseased cell becomes an indispensable need.

Stimuli-responsive nanocarriers including liposomes, polymeric micelles, polyplexes and lipoplexes are having great importance in considering this situation of triggered release of drugs [6,7]. They are capable of responding to environmental changes like temperature, pH, biomaterials, solvent, ionic strength, light, chemical agents, electrical fields and magnetic fields [8,9,10]. Among these pH-responsive nanocarriers have unique importance for the triggered release of drugs at the targeted sites by responding to the different pHs in various parts of the body [11]. The main disadvantage of conventional chemotherapy is that it cannot apply the drug specifically to the cancer cells and thus leads to damage of normal living cells. The extracellular pH is 7.4, while in the tumor cells the pH is 5–6 because of the rapid and frequent cell proliferation and lactic acid accumulation [12]. A pH-responsive carrier molecule that can respond only to acidic pH might be an efficient way to release drugs precisely in the tumor environment causing precise cytotoxicity to the cancer cells. The gastrointestinal tract, stomach, duodenum, jejunum, ileum and vagina have different pHs, which is suitable for ensuring the desired level of drug uptake at the respective site [13,14,15,16,17]. Thus, controlling the nanocarrier’s properties such that it can release the drug in a steady manner that can maintain the concentration of drug at the desired site for a long time is useful to avoid the frequent intake of medicines. In this way, it is possible to avoid the inconvenience of patients by decreasing the chance of destruction of normal living cells along with an effective way of killing the tumor cells.

Various polymeric materials have been designed with reasonable pH-responsive properties, improving the efficacy of drug release rates with respect to the pH variation. Among them, their biodegradability, biocompatibility, stimuli-responsive nature and the ability to form micelles have made polypeptides some of the best candidates for the synthesis of carrier molecules for biological systems [18,19]. The backbone of polypeptides consists of peptide linkages between amino acids, that can donate and accept hydrogen bonds. The use of different amino acids (AAs) in the polymeric main chain or side chains will contribute significant features to the final polypeptides. The linear sequence of AAs in the polypeptides is referred as the primary structure, which finally determines the three-dimensional shape of the polymer. Secondary structures like α helices, β sheets, β turns and ω loops are formed by the hydrogen bonding between the -NH- groups and carbonyl group of the peptide backbone. Depending on the AAs’ structure, the preference of the secondary structure tends to be different. AAs like lysine and glutamate prefer the formation of α helices while proline is known as a helices breaker due to its effects on the breaking the backbone conformational regularity of helical chains. Proline usually prefers the β turn structure. The aromatic bulky groups in the side chain of tyrosine and tryptophan leads to the selective formation of β strands in the β sheet. Although this preference is not useful for predicting the structure, the hydrogen bonding interactions can be influenced by the pH, temperature, and other environmental stimuli. Hence regulating the stimuli, the conformation can be altered and subsequent control on the biological activity of the polypeptides can be achieved. The design of zwitterionic nanomaterials has also been achieved by utilizing the basic AAs (lysine, arginine and histidine) or acidic AAs like glutamate and aspartate. In this review, we focus on the recently reported pH-responsive polypeptide-based nanocarriers for medical applications. Furthermore, we discuss drug delivery at the organ, tissue and cell level thus adding value to recent advancements in the pH-responsive polypeptides including the use of other responsive carriers with pH-responsive peptides to achieve a better efficacy for therapeutic applications.

## 2. Types of pH-Responsive Polypeptides

Polypeptides have been widely used in the development of the nanocarriers for theranostic applications mainly because of their biocompatibility. The pH-responsive nature of some AAs confers an extra benefit in biological applications, especially in drug delivery.

### 2.1. Cationic Polypeptides

Cationic polypeptides consist of AAs like lysine, histidine, arginine, and in few cases proline, tyrosine and tryptophan or derivatives of these AAs. These polypeptides are featured by the capability of deliver the molecule into mammalian cells by elucidating a pathway for internalization [20]. Hence, they are known as protein transduction domains (PTDs). Even though PTDs have attracted much attention, their mechanism of internalization remains controversial [21]. In one situation cytosolic penetration likely involves direct plasma penetration and in another situation a two-step process of endocytosis followed by escape from the endosomes were also observed [22,23,24,25,26,27,28,29]. The cell fixation alters the transduction of cationic peptides and internalization is an energy-dependent process [30,31]. The use of these peptides has been attempted for the delivery of drugs, genes, peptides and proteins in targeted sites [32,33,34,35].

#### 2.1.1. Poly(l-lysine)-Based Polypeptides

Lysine is an essential AA that cannot be synthesized in our body and must be obtained from the diet in the form of lysine or lysine-containing proteins. Lysine has been investigated for prevention and treatment of cold sores, herpes infections, and osteoporosis because it is helpful at preventing the loss of calcium by absorption in the intestine. Nevertheless, lysine intake from food materials is limited due to the loss during various food preparation methods. The pH-responsive nature and biocompatibility of lysine along with its requirement in the body have led its use in medical treatment, especially in drug delivery applications [36,37,38,39]. Specifically, poly(l-lysine) (PLL) and pseudo-PLLs has been used for different kinds of pH-responsive nanocarriers. There are two kinds of PLL: i.e. α-PLL that is synthesized by artificial means and shows intrinsic toxicity, and ε-PLL that is synthesized using microbial means, having wide application in medicinal, electronics and food [40,41,42,43].

In 2005, Yue and coworkers demonstrated a pseudopeptide based on poly(l-lysine iso-phthalamide) grafted onto the poly(ethylene glycol) (PEG) analogue, Jeffamine M-1000. The conformational transition depending on the degree of grafting had a great effect on the pH-response and the pH-range, because the grafting was capable of encouraging hydrophobic associations of polymer backbones and modifying the onset of conformational changes. Below 35.1 wt% grafting, the product retained the pH-induced conformational transition of the parent polymer due to a compact hydrophobically stabilized structure at low degree of ionization [44]. The high degree of grafting of poly(l-lysine isophthalamide) significantly enhanced the therapeutic efficiency by pH-dependent change in hydrodynamic size. Ho et al. proposed pH-responsive endosomolytic pseudopeptides obtained by grafting l-phenylalanine onto the poly(l-lysine isophthalamide). The hydrodynamic size of polymer decreased by lowering pH owing to the increase of hydrophobicity and the protonation of acidic groups, resulting in a more compact conformation. The relatively small size of the polymer helps achieve the exterior to interior penetration and cell internalization. Hence, this biomimetic pH-responsive and biodegradable pseudopolypeptide has significant therapeutic delivery ability [45]. The selective release of drugs at a specific site depending on the suitable range of acidic pH values was also accomplished using lysine-based nanoparticle (NP) systems. Nguyen et al. presented a drug delivery system based on 3-aminopropyl-functionalised mesoporous silica NPs (MCM-NH_2_) coated with succinylated ε-PLL (SPL). The SPL acted as a nanogate to precisely release prednisolone in the environment of the colon (pH 5.5–7.4), but it shows no release in the small intestine (pH 5.0) or stomach (pH 1.9). In addition, the SPL-coated NPs could deliver the cell membrane impermeable dye, sulforhodamine B, to the intestinal epithelial cancer cells and RAW 267.7 macrophages (Figure 1). These are promising drug delivery systems for colon diseases [46].

A two-component toothbrush-type superamphiphile based on PLL was recently described for the pH-responsive assembly and disassembly of spherical aggregates. One fragment is a double-hydrophilic block copolymer, methoxy PEG-*b*-PLL hydrochloride, and the other fragment contains 4-(decyloxy)benzaldehyde. The amine group of lysine and the aldehyde group of 4-(decyloxy)-benzaldehyde form an imine bond under physiological pH (7.4). When the pH reaches 6.5, the benzoic imine bond dissociates to decompose the superamphiphiles. Thus, the self-assembly and disassembly of the aggregates occur depending on pH [47]. The NP containers fabricated by A PLL-pyranine-3 showed dynamic pH-responsive sensing characters.

Spherical NPs of coacervated PLL-pyranine-3 were prepared by the counter-ion condensation of PLL in the presence of pyranine-3. Finally, the combination of pH-responsive release with the fluorescence of pyranine-3 in the microcapsule structure suggested a hopeful strategy for the spatially and temporally controlled release of the guest molecule [48]. Naik et al. reported a new lysine block polymer system, poly(propylene oxide)-*b*-PLL, exhibiting significant shift in the pH associated with helix-coil transition of the PLL block [49]. The resultant vesicle morphology showed pH and temperature-responsive drug release. The alginate beads coated with basic PLL exhibited a controlled release of fluorescein isothiocyanate (FITC)-dextran in respond to change in pH. The peptide linkage of lysine was more protonated at lower pH, resulting in a lowering of the degree of release around pH 3.5 and pH 4.0 by suppressing a swelling of alginate [50].

A star-block copolymer has been synthesized as a potential nanocarrier for the release of anionic drugs in two different pH ranges. The PLL block of the inner shell and the PEG of outer shell are grafted onto the PEI core, PEI-(PLL-*b*-PEG). The peripheral primary amine groups of PEI initiated the ring-opening polymerization of ε-benzyloxycarbonyl-l-lysine *N*-carboxyanhydride. The surface was then modified with the activated PEG 4-nitrophenyl carbonate. This star-block copolymer could efficiently encapsulate the model drug, diclofenac sodium and anionic dyes such as methyl orange and rose Bengal and released at the low pH 2.0–3.0 and at the high pH 10.0–11.0 [51]. A pH-responsive comb-shaped copolymer NP was also demonstrated to improve the protein stability in the harsh internal pH environment. Gao et al. designed an amphiphilic copolymer, lactobionic acid-PEG-*b*-PLL-*g*-poly(lactide-*co*-glycolide) (LPEP) [52]. The internal pH of LPEP was adjustable depending on the ratio of poly(lactide-*co*-glycolide). The stability of proteins released from LPEP was confirmed by circular dichroism spectroscopy and was better than that released from poly(lactide-*co*-glycolide) NPs. Liu et al. reported a tetrablock polymer nanocarrier for pH-responsive anti-cancer drug delivery [53]. Nano-micelles designed based on poly(l-lysine-4-azepan-1-yl-butyric)-*b*-PEG-*b*-poly(leucine) were able to response to different pHs. Chemical cross-linking of PLL with ethylene glycol diglycidyl ether yields a pH-responsive hydrogel, which can change its volume and conformation from α-helix to random coil with change in pH (Figure 2) [54]. Recently a pH-responsive asymmetric dendron-like polypeptide-based polyampholyte was reported by Chen et al. Poly(l-lysine)_4_-d_2_-*b*-d_1_-poly(l-glutamic acid)_2_ [(PLL)_4_-d_2_-*b*-d_1_-(PLGA)_2_], where D_1_ and D_2_ are the first and the second generation poly(amido amine), was synthesized by the combination of ring-opening polymerization and click chemistry. The (PLL)_4_-d_2_-*b*-d_1_-(PLGA)_2_ showed schizophrenic micellization behaviors (Figure 3) and self-assembled into aggregates as PLL-corona and PLGA core in acidic pH, while PLGA corona and PLL core in alkaline pH due to coil to helix conformational transition. Moreover, the self-assembled aggregates changed their morphologies from spherical to rod-like and then to spindle-like micelles [55].

So-called pH (low) insertion peptides (pHLIPs) that can be exploited as biomarkers for targeting imaging agents and precise transport of drugs to cells in acidic diseased tissues have been used as a specific ligand for targeting the tumor acidic environment by differentiating between the pH of healthy and diseased cells, especially to detect and diagnose tumors at the early and metastatic stages [56]. The PLL modified with pHLIP formed nanoclusters with superparamagnetic iron oxide (SPION). To target the acidic cancer microenvironment at the initial and metastatic stages, pHLIP was utilized as a precise ligand. SPION are contrast enhancing agents utilized in noninvasive magnetic resonance imaging for cancer cells. This exceptional contrast improvement of the cancer inner core by the pH-responsive SPION in three various tumor models verified the clinical efficiency of this NP. PLL was also utilized to develop pH-responsive single walled carbon nanotube (SWNT) dispersions [57]. This environment-responsive polymer-SWNT complex has promising applications in sensing, nanoelectronics, and gene and drug delivery systems. Since SWCNTs are highly hydrophobic and typically aggregated in bundles, it is necessary to control the aggregation and dispersion of SWCNTs in aqueous solution with external stimuli using pH through PLL [58].

Lysine-based polymer NPs are successfully demonstrated for protein delivery. Liu et al. demonstrated the delivery of protein (bovine serum albumin – BSA) using a pH-responsive comb-shaped block copolymer, PEG monomethyl ether-*b*-PLL-*g*-poly(lactic acid). Usually the harsh pH level inside the polymer carriers restricts the delivery of protein. Interestingly this polymer could automatically adjust the internal pH to a milder level, which helps the NPs to achieve a sustained protein release under both in vitro and in vivo conditions [59]. PLL itself lacks in nucleic acid transfer activity, however, modification with suitable functionalization can show high efficiency in siRNA delivery. Polycations having nucleic acid binding, compaction, protection and biocompatibility are suitable for siRNA delivery [60]. A pH-responsive gene carrier prepared by coupling the amine groups of linear poly(allylaimine) (PAA) with PLL also formed stable complexes with DNA and the resultant pH-responsive α-amino groups enhanced the transfection activity [61]. The zwitterionic triblock copolymer, polyethylenimine-*b*-PLL-*b*-poly(l-glutamic acid), with negative charge at physiological pH could act as an effective system to shield the positive charged polyplexes [62]. In extracellular environment, tumor cell pH is about 6.5, thus, the positive charges of polyplexes could be restored in tumor cells. This is advantageous for the enhanced cellular uptake and the better transfection efficacy due to the activated electrostatic interaction between negative cancer cells and positive polyplexes.

The grafting of some molecules to the pH-responsive polymer significantly enhances the therapeutic efficiency by pH-dependent change in hydrodynamic size. For example, gene delivery capability can be significantly enhanced by modifying the cationic polypeptides to functional anionic polymer. Ternary complexes obtained by coating of *cis*-aconite anhydride and functional peptide grafted PLL onto binary gene complexes exhibited enhanced gene delivery efficiency at the mRNA and protein level gene expression level [63]. Co-delivery of drug and osteogenic gene for osteo-differentiation has been demonstrated using PLL-modified polyethylenimine copolymer and arginine-glycine-aspartate peptide were successfully anchored onto the surface of mesoporous silica NPs [64]. Polymeric micelles assembled by stereocomplexation of poly(l-lactic acid)-*b*-PL/poly(d-lactic acid)-*b*-methoxy PEG exhibited slower drug release and weaker efficacy of intracellular proliferation inhibition than normal polymeric micelles [65]. Oriented microtube structures containing PLL-heparin sodium NPs have also been investigated for the controlled release of TGF-β1in cartilage tissue engineering. These scaffolds exhibited optimal properties for the controlled release and excellent cartilage repair effects [66].

The nanomicelles fabricated from a PLL-based biodegradable polypeptide hybrid terpolymer exhibited stepwise pH- and redox response and pH-triggered charge reversal property for endosome escape [67]. Biocompatible, hemocompatible, and highly sensitive antimicrobial PLL-based peptidopolysaccharides were also reported. The cationic polypeptide was prepared by thiol-ene click chemistry of thiolated polysaccharide backbone and the antimicrobial polypeptides, methacrylate-ended poly(lysine-*co*-phenylalanine). This hybrid material displayed 200 times higher selectivity than polypeptide molecules [68]. Furthermore, PLL containing calcium phosphate particles can be utilized for highly sensitive glucose detection. The glucose oxidase immobilized on PLL bearing calcium particles displayed 80% higher oxidation activity compared to that of native glucose oxidase and maintained approximately 70% activity even after ten cycles [69]. The PLL-based nano-probes were reported as novel ultra-sensitive electrochemiluminescence biosenser for glutathione (GSH), in which the PLL was used as co-reactant of luminol and poly(luminol/aniline) nanorods loaded onto reduced graphene oxide [70].

#### 2.1.2. Poly(l-histidine)-Based Polypeptides

Poly(l-histidine) (pHis) is another important pH-responsive cationic peptide derived from one of the 22 proteinogenic AAs and bears an imidazole functional group. Just like other imidazole compounds, histidine also shows anti-oxidant, anti-inflammatory and anti-secretory properties. In addition, the scavenging capacity of the reactive oxygen species induced by the imidazole ring contributes efficacy in protecting inflamed tissues. His is also essential for blood cell prparation and to prevent tissues damage caused by heavy metals and radiation. Considering these features and its biocompatibility along with the pH-responsive features, pH is widely used in theranostic application.

Poly(lactic acid)-*b*-PEG-*b*-pHis formed flower-like micelles that can be used as a tumor pH-responsive anticancer drug carrier. In the flower-like assembly, poly(lactic acid) (PLA) and pHis blocks form the core and PEG block forms the shell. The ionization of pHis at slightly acidic pH leads to the micellar core deformation (Figure 4) and is responsible for the triggered drug release from the micelles with respect to small changes in pH [71]. pHis was utilized to confirm the pH-triggered shielding and deshielding approach by modulating the properties of cationic micelles. In this way the paclitaxel release kinetics, stability in plasma, in vitro cellular uptake and in vivo infiltration and anticancer efficiency could be tuned.

The surface of cationic micelles assembled from a block polymer of pHis and short branched polyethyleneimine was shielded by electrostatic complexing with a negatively charged PEG methyl ether (mPEG)-*b*-polysulfadimethoxine. The cationic NPs showed a unique pH-responsive mechanism for cancer therapy [72]. Thermo- and pH-responsive block copolymers composed of PEG, poly(*d*,*l*-lactide-*co*-glycolide) (PLGA) and pHis were synthesized to use as stimuli-responsive nano-carrier micelles for drug. Firstly PLGA-*b*-PEG-*b*-PLGA was prepared by the ring opening polymerization (ROP) of *d*,*l*-lactide and glycolide using the PEG as a initiator. Then pHis prepared by the ROP of His *N*-carboxyanhydride using isopropylamine as an initiator was coupled with PLGA-*b*-PEG-*b*-PLGA. The two hydrophobic blocks, PLGA and pHis, form the core of the micelle and the hydrophilic PEG formed the shell [73].

The conjugation of the reductive end of dextran to the amine group of pHis gave a dextran-*b*-pHis that assembles to form pH-responsive NPs [74]. The particle size increased in acidic pH and decreased in basic pH. The anticancer drug, doxorubicin (Dox) loaded in the NP using nanoprecipitation dialysis method released more actively at acidic pH. Hyaluronic acid (HA) conjugated with pHis was also reported as the stimuli-responsive micelles [75]. Studies of particles of different size confirmed the pH-responsive behavior of HA-pHis conjugate micelles. The HA-pHis micelles exhibit the highest degree of cellular uptake at the lowest degree of substitution. The Dox-loaded polysebacic anhydride nano-carriers coated with pHis were also studied for the pH-responsive nano-carriers [76]. Interestingly nano-carriers covered with PEG reduced the macrophage uptake.

Dual stimuli-responsive vesicular nanospheres were fabricated using pHis and phospholipids as tumor targeted photodynamic therapy [77]. In this work, the smart nano-spheres were fabricated by the simple self-assembly of the mixture of the soybean lecithin-derived phosphatidylcholine, phosphatidylethanolamine-pHis conjugate and the folic acid-conjugated phosphatidylethanolamine-poly(*N*-isopropylacrylamide) (pNIPAm). The vesicular nanospheres encapsulated with chlorin e6 (Ce6) showed a significant photodynamic therapeutic efficiency (Figure 5) on KB cells than pure Ce6. Using pHis, a series of pH/redox dual stimuli-responsive zwitterionic micelles also recently developed for intracellular delivery of Dox into tumor cells [78], where pHis was combined with a biocompatible phospholipid analog poly(methacryloyloxyethyl phosphorylcholine) using a redox-responsive disulfide linker. Very recently, another series of dual stimuli-responsive pHis-based triblock nano-carriers have been developed for intracellular tumor-targeted drug delivery [79]. In this work nano-carriers were fabricated from the two pH-responsive pHis end-blocks and polyurethane as middle-block, tethered by a redox-responsive disulfide linker. The in vitro drug release profile showed an enhanced Dox release in an acidic environment in the presence of 10 mM GSH. Using the pHis as the pH-responsive core, multi-responsive disulfide cross-linked polypeptide nanogels were produced by a one-step ROP. In this work, the corona of the nanogel was used as the reductively cleavable cross-linking agent [80].

A triblock copolymer based on PLL and pHis was used to enhance the airway gene transfer by DNA NPs. The pH-responsive DNA NP formulated to mediate gene delivery via nucleolin-independent pathway. Inclusion of pHis block between PEG and PLL increased the buffering capacity of final polymer. Moreover, the triblock PEG-*b*-pHis-*b*-PLL DNA NP enhanced the in vitro gene delivery by about 20-fold and in vivo gene delivery to lung airways in BALB/c mice by about 3-fold. The highly compacted pH-responsive DNA NPs mediated transgene silencing have been reported by Kim et al. [81]. These NPs efficiently silenced a tumor-specific transgene (firefly luciferase). Ce6 was encapsulated in the multi-functional hybrid NPs assembled by using HA and pHis conjugate linked by disulfide bond. The NPs was used for the diagnostic and drug release applications. The NPs showed specific targetability and tumor cell growth inhibition in CD44 receptor in a pH-responsive manner [82].

A dual-responsive pH-sensor using pHis block was also reported [83]. The pHis quencher was linked with Atto488-labeled nitrilotriacetic acid fluorophore by a coordination with cobalt(II) ion. The resulting polymer was sensitive to both acidic and basic conditions. The multi-functional polymeric NPs fabricated by PEG-*b*-pHis-*b*-d-a-vitamin E succinate copolymer were used for reversing tumor multidrug resistance. The triblock copolymers afterwards functionalized with biotin for targeted delivery, exhibiting a remarkable impact on the obstruction of P-gp ATPase activity of MCF-/ADR cells, the reduction of intracellular ATP level and, the loss of mitochondrial membrane potential [84]. The use of pHis-modified liposomes as a cell penetrating vehicle to deliver the large cargos such as α-galactoseidase A, a lysosomal enzyme to the intracellular lysosomes was also attempted [85]. The hydrogel based on multivalent coordination of Ni^2+^ with pHis-terminated PEG and multiple iminodiacetic acid-modified oligochitosan that shows enhanced neutral stability was used as a pH-responsive self-healing system [86]. pHis block could be used to fabricate NPs with intelligent functions along with the pH-responsive drug delivery. In the normal physiological medium, pHis close the active targeting function and expose only in acid tumor tissue [87]. The methoxy PEG-*b*-polylactide-*b*-pHis-ss-oligoethylenimine copolymer formed a dual-responsive polyplex with effective endo-lysosomal escape. The codelivery of siRNA and Dox from the polyplexes showed highly efficient tumor growth inhibition and downregulation of P-gp expression of the MCF-7/ADR cell [88].

#### 2.1.3. Polyarginine-Based Polypeptides

Arginine-rich peptides and polyarginine (pArg) are considered as an efficient protein transduction domain (PTD) conjointly known as cell-penetrating peptides (CPPs) that are better than Lys or His. Hence, the pArg-based polypeptides are mainly designed for better overcoming the cellular barrier. They degrade upon incubation with peptidases and other endogenous enzymes [89]. The biodegradability of the pArg has been confirmed by number of convincing reports [90,91,92].

Self-assembled polymeric vesicles are of high potential as robust encapsulants. Contrasted with liposomes, polymeric vesicles have many advantages, along with the introduction of biofunctionality facilitating their interactions with cells and tissues. The polymeric vesicles fabricated using pArg and polyleucine (pLeu) segments could entrap ester soluble species and could be switched to various sizes and with good yield. pArg fragments derive vesicle formation and offer functionality for effective intracellular delivery [93]. The effect of pArg on the ocular absorption of hydrophilic molecule after instillation was studied by Nemoto et al. [94]. pArg did not make any alteration on neutrophil infiltration, disruption of the epithelial and stromal morphologies, production of TNF-a and corneal epithelial and stromal thickness.

The complexation of the pArg to the low molecular weight (MW) heparin enhanced the pulmonary absorption of the anionic drug and reduced the epithelial toxicity. The pulmonary formulation of enoxaparin plus 0.0125% or 0.0625% of pArg led to a two-fold rise in the relative bioavailability of low MW heparin compared to the enoxaparin plus normal saline [95]. pArg was also used for the transport of the *N*-terminal AAs AVPIAQK (SmacN7). The expected strong nuclear uptake of pArg containing SmacN7 conjugate mediated cell death in tumor and healthy cells [96]. Ternary complex microcapsules of alginate/pArg-chitosan were fabricated by coating chitosan and pArg as membrane materials on calcium alginate beads [97]. The investigation of the in vitro drug release alginate/pArg-chitosan microcapsules exhibit more sustained and stable macromolecular drug release compared with alginate/chitosan microcapsules. Within 85 h, the microcapsules released 85.7% of the bovine erythrocytes hemoglobin (Hb) in pH 6.8 PBS. But in a pH 1.0 HCl solution, only 9.6% Hb release was obtained in the first half hour and then it became unchanged, indicating that the alginate/pArg-chitosan microcapsules possess a pH-response property. Semipermeable membrane made of pArg and dextran sulfate formed the self-exploding microcapsules for pulsed drug delivery. The microcapsules comprise of a microgel based on biodegradable dextran encircled by the pArg and dextran sulfate-based semipermeable membrane. The resultant microcapsules were capable of releasing the drug under physiological conditions in a pulsated fashion [98].

A non-covalent co-incubation strategy was used for the nucleic acid delivery. Cell-penetrating peptides pArg and its derivatives have been readily used for nucleic acid delivery. Arginine-rich peptides frequently remain trapped in endosomal compartments following internalization. This problem could be solved by conjugating a stearyl moiety to pArg [99]. Lemeshko et al. suggested that pArg cell delivery vectors of antitumor polycationic peptides increased their direct potential-dependent penetration of biological membranes and created the capacity to cause mitochondrial aggregation [100]. Shah et al. investigated the pArg chain length influence on the delivery of surface-modified nanostructured-lipid carriers (NLCs). A hot-melt technique was used to prepare the NLCs and its surface was then modified with six-histidine tagged cell penetrating peptides. The number of Arg required for deliver active drug into deeper skin layers was optimized using the design of experiment (DOE) [101]. Follicular and non-follicular pathways for pArg and surface modified cell penetrating peptide NP are also analyzed, demonstrating that the increased skin permeation of surface modified NPs was dependent on follicular pathway and occur through non-follicular pathway [102].

Simple mixing aqueous solutions of pArg and polysaccharide HA at room temperature yielded nanocarriers of particle sizes ranging from 116 to 155 nm and zeta potentials ranging from −31.3 to −35.9 mV. The pArg-HA complex prepared with high MW HA formed spherical morphologies and remained stable after isolation by centrifugation and in physiological medium [103]. The pArg nano-capsules modified with use of an oily core and stabilized with phosphatidylcholine was used to overcome cellular barriers. These nano-capsules consisted of pArg corona that can accommodate polar negative charged ions such as plasmid DNA and an oliy core that can accommodate lipophilic drugs. The nanocapsules of 120–160 nm diameter and +56 zeta potential quickly and massively accumulated in NCI-H460 cells. The PArg shell played a critical role in the internalization process. The docetaxel (DTX)-loaded nano-capsules inhibited proliferation efficiently compared to free drug. pArg nano-capsules were successfully administrated as carriers for the oral delivery of peptide drugs [104]. An antitumoral peptide, elisidepsin was effectively loaded into the nano-capsules by adjusting the formulation parameters.

A giant unilamellar vesicle was used to study numerically and experimentally the interaction mechanism between pArg and lipid bilayer. This work provided a broad perception for the peptide infiltration mechanism and proposals for the gene and drug delivery [105]. The cytosolic penetration is often context dependent. Wang et al. reported the oxidation state of cell-dependent cytosolic penetration of pArg. Hypoxia and cellular antioxidants inhibited the cell penetration. At the same time, the oxidants promoted an efficient cytosolic cell entry comparative to the membrane generating reactive oxygen species level [106]. One more study confirmed the cell penetrating nature of polymeric micelles with pArg decoration. Five kinds of reduction-degradable polyamide amine-*g*-PEG/pArg (PAA-g-PEG/pArg) micelles with various proportions of hydrophilic and hydrophobic blocks were fabricated. The remarkable biocompatibility of the PAA-g-PEG/pArg micelles was demonstrated by cell cytotoxicity experiments and they were effectively internalized into human hepatocellular carcinoma (HepG2), inhibiting the cell proliferation [107]. pArg nanocapsules were also used to overcome the intestinal barriers during oral administration. The nanocapsules composed of an oily core and a pArg shell with insulin as model peptide showed enhanced penetration properties. After several formulation trials, pArg combined with oleic acid, sodium deoxycholate and Span 80 proved to be optimal and the resultant nano-capsules of an average size of 180 nm with low polydispersity exhibited high insulin association efficacy, good colloidal stability upon incubation in simulated intestinal fluids and a good capacity for the controlled release of insulin [108].

pArg nano-capsules can be used for transmucosal drug delivery. The nanocapsules with an oily core fabricated with a single pArg or double pArg/polyacrylic acid layer demonstrated a reasonable encapsulation of the hydrophilic model peptide salmon calcitonin and controlled release in simulated intestinal fluids. Furthermore, the nanocapsules reduced the transepithelial electrical resistance of the monolayer without reducing the viability during Caco-2 epithelial cell line [109]. pArg conjugated PEG-lipid has been used for gene therapy using small interfering RNA delivery. pArg conjugation helped to overcome the limitations such as insufficient cellular uptake and poor stability which limited the use of siRNA delivery. Moreover, the novel PEG-lipid cationic liposome formulation demonstrated improved intracellular siRNA delivery and reduced H4II-E and HepG2 cell line cytotoxicity, and it effectively reduced the GFP protein expression levels of gene [110]. Zhao et al. demonstrated a safe and efficient siRNA delivery system using a biodegradable amphiphilic tri-block copolymer, cationic monomethoxy PEG-*b*-poly(*d,l*-lactide)-*b*-pArg for safe and efficient siRNA delivery. The self-assembly of the polymer led to the cationic polymeric nano-micellles with 54.3 nm diameter and 34.8 mV of zeta potential, exhibiting greater cell viability and hemocompatibility [111].

A prostate cancer-specific targeting nano-carrier system was studied for a delivery of microRNA [112]. The nanomaterials fabricated using the pArg_11_-labeled branched polyethyleneimine bearing non-toxic disulfide (−S–S−) linkages (pArg_11_-SSPEI) displayed full bioactivity in completely nontoxic situations. The mesoporous bioactive glass (MBG) NPs with polyglycerol coating were conjugated with pArg_8_ for DNA delivery [113]. The MBG-pArg_8_ NPs prepared via a sacrificial liquid template method in sol-gel process showed higher plasmid DNA loading and cell transfection efficiency than mPEG-modified with amino groups. The inhibition system of *N*-methyl-d-aspartate-induced retinal neuronal death could be made using pArg [114]. The pArg-rich transport moiety linked to the PDZ-PSD-95-binding cyclic peptide was adequate to facilitate long- and short-term protection through a mitochondrial targeting mechanism. The pArg_11_ combined with hydrophobic counter anion 4-(1-pyrenyl)-butyric acid (PB) could be applied for protein transduction in transdermal delivery [115].

Movafegh et al. reported that the pArg fragments at the minimum and at the maximum concentration act as proliferation inducing and antiproliferative agents [116]. In between the pArg increased the cellular uptake of Dox and necrotic cell death. Junmei et al. showed that pArg penetrated the membrane through a water pore on the membrane and by conjugating to small NPs with proper linkers, its transmembrane efficacy could be enhanced. The length of the linker around half of the membrane thickness was proper to get the maximum translocation efficiency [117]. Transdermal delivery of triptolide has been reported with C-14-hydroxyl group of triptolide modified with pArg [118]. The MTT assay on the immortal human keratinocyte (HaCaT) cell line showed that the system has a high potential as a topical therapy by transdermal delivery of triptolide.

#### 2.1.4. Polytyrosine-Based Polypeptides

Tyrosine is a hydrophilic, non-essential AA with a special role in proteins for signal transduction processes. The oral ingestion of tyrosine is effective to exert acute effects on catecholamine systems within and outside of the brain [119]. It is useful in stress relieving by reduction in stress hormone levels [120]. Tyrosine is a principal AA for the radioiodination of proteins or other molecules.

The polytyrosine (pTyr) has an effect on the modification of surface hydrophobicity. pTyr was incorporated into hydroxyethyl starch microspheres to permit radioiodination and assessment of uptake into intestinal mucosa during oral drug administration [121]. The hydrophobicity of the microspheres was evaluated by water uptake of the microsphere and contact angle examination using sessile drop method. The water uptake was significantly reduced by incorporating pTyr. The contact angle of pure hydroxyethyl starch films was 16°, but the value increased to 29°, 39° and 57° by incorporating 5%, 10% and 20% of pTyr, respectively. Recently an article described the enzyme-catalyzed green synthesis of an unnatural poly(amino acid) (pAA) with zwitterionic character and pH-responsive solution behavior [122]. The tyrosine was polymerized using linear-dendritic laccase complexes as initiators and water as solvent. pTyr with MW up to 82 kDA was obtained in the relative composition of pTyr ranging between 45 and 69%. The final product has a resemblance with the living polymer, so that chain grew further upon addition of fresh monomer. The resultant polymers were soluble in water and behaved as poly(zwitterion)s with their size and charge varying according to pH.

Lewandowski et al. reported a chitosan-based pro-tag for the enzyme-catalyzed protein capture and release [123]. For the capture of protein tyrosinase that oxidizes available tyrosine residues was used. Enzyme activation essential for protein capture by chitosan was enhanced by engineering the protein to have a pTyr_5_ fusion. The captured GFP from chitosan conjugate was released using two enzymatic methods. Firstly, enterokinase was utilized for the protein cleavage at an engineered EK-cleavage site and secondly, chitosanase was used to hydrolyze the chitosan backbone. A reductive responsive, small and robust polypeptide micelles based on cyclic arginine (Arg)-glycine (Gly)-aspartic (Asp) peptide (cRGD)-decorated pTyr showed a high drug loading content of 18.5 wt% and remarkably suppressed the growth of MDA-MB-231 human breast tumor without inducing side effects [124]. Recently, a cRGD-decorated pTyr-based NPs was developed for robust encapsulation and targeted Dox delivery to colorectal cancer [125]. These NPs boosted the encapsulation and showed an ultra-high Dox encapsulation with drug loading contents ranging from 18.5 wt% to 54.1 wt%. Moreover, the Dox-loaded NPs showed an enhanced stability against dilution, serum and Triton X-100 surfactant and released the payloads quickly in cancer cells. Different micelles formed by the self-assembly of cationic polypeptide-based polymers are summarized in Table 1.

#### 2.1.5. Polyproline- and Polytryptophan-Based Polypeptides

Proline is a non-essential, aliphatic and non-polar AA that is synthesized in the body from *l*-glutamate. It is the only proteogenic AA with an α-amino group attached to the side chain. Proline have exceptional rigidity because of its characteristic cyclic structure. It has three important consequences due to this unusual nature in AA: (1) the backbone conformation of proline is restricted, (2) the bulkiness of the *N*-CH_2_ group places restrictions on the conformation of residue proceeding proline and (3) it is unable to act as hydrogen bond donor due to the amide proton is replaced by CH_2_ group [126]. The main characteristics of the poly(l-proline) (pPro) is that its tertiary amide group leads to significant lowering of the barrier for *cis-trans* amide isomerization [127]. Further, the l-proline-based nanogel reported as a successful pH-switcher [128]. The recent report based on pPro tri-helix macrocycles as nanosized scaffolds suggested that they were capable to probe the rearrangement of receptors on the cell surface and manipulated signalling or cell recognition as nanomedicine [129].

Tryptophan is an essential AA containing indole as side chain that contributes to the pH-responsive nature. A graft copolypeptide containing tryptophan was reported for the release of hydrophobic drugs [130]. The amphiphilic copolypeptide was synthesized by an aminolysis reaction of poly(γ-benzyl-l-glutamate)-*graft*-poly(l-tryptophan) with amino-1-ethanol followed by treatment with trifluoroacetic acid. The hydrophobic drug, testosterone was successfully encapsulated in the hydrophobic moiety and was subsequently released at very low pH (2.0); however, the release was depressed at pH 5.0 due to the aggregation and dissociation of tryptophan in the graft copolypeptides. Similar behavior was observed for the amphiphilic copolypeptide, poly(*N*-hydroxyethyl-l-glutamine)-*b*-poly(l-tryptophan) [131]. The release behavior of 8-anilino-naphthalene-1-sulfonate (ANS) and pyrene studied at various pH showed that the trypotophan residue dissociated in faster rate at pH 2.0 than at pH 5.0 and confirmed that the particle size of nano-micelles changed from 90 nm at pH 5.0 to 50 nm at pH 2.0. Various pH-responsive drug release systems based on cationic polypeptides are summarized in Table 2.

### 2.2. Anionic Polypeptides

Anionic peptides consist of acidic functional groups as pendants in their side chain and show negative charge due to the deprotonation in physiological pH. Aspartic acid (Asp) and glutamic acid (Glu) are the significant negative AAs that can be used for making biocompatible polypeptides for pH-responsive carrier systems. The pH-responsive nature of the polymer towards the external pH depends on the number groups of negative charges on the polymer [132,133,134,135,136]. Polyacids usually undergo ionization depending on their pK_a_ value. The pK_a_ value of Glu and Asp are 4.07 and 3.90, respectively [137].

#### 2.2.1. Poly(aspartic acid)-Based Polypeptides

Poly(aspartic acid) (pAsp) is a water-soluble and biocompatible polypeptide that has wide applications for agriculture, mineralization, corrosion inhibitor, anti-toxics, super-swelling agent in feminine hygiene products, food packaging and medicinal field [138,139,140]. An amphiphilic multi-arm block copolymer conjugated with Dox via a pH-responsive linker was developed using pAsp as one block (Figure 6). Dox was covalently conjugated via a pH-responsive hydrazone linkage in a pAsp block. The release profile of the Dox from the micelle demonstrated a strong pH dependence because of the acid cleavable hydrazone linkage at tumor pH [141].

A salt, pH and temperature responsive semi-interpenetrating hydrogel was synthesized using pAsp and poly(acrylic acid). This multi-responsive hydrogel was ionic in nature, had a polyelectrolyte complex structure, and the medium significantly influenced the swelling behavior. The incorporation of pAsp improved the responsive behavior of this hydrogel, alternately changing by inorganic salt, pH and temperature along with suitable mechanical strength during repeatable swelling and shrinking period [142]. pAsp was used along with xanthan gum to synthesize pH, temperature and ionic strength responsive interpenetrating polymer network (IPN) hydrogels [143]. The pAsp feed composition was a significant factor to tune the IPN hydrogel properties. A biodegradable dual stimuli-responsive amphiphilic hydrogel was prepared using pAsp derivative, poly(*N*-2-hydroxyethyl-d,l-aspartamide). Up to that time, this derivative was used only for making pH-responsive hydrogel [144]. Moon et al. proposed temperature- and pH-responsive amphiphilic hydrogels fabricated by successive aminolysis reactions of polysuccinimide (PSI) using both hydrophobic *N*-alkylamine and hydrophilic *N*-isopropylethylenediamine. These injectable double responsive hydrogels showed potentials for tissue engineering and drug delivery [145]. A similar dual responsive IPN hydrogel was prepared using pAsp with pNIPAm. The IPN hydrogel had large and uneven porous network, different from usual pNIPAm hydrogel structure. In addition to this, it showed faster shrinking and re-swelling depending on relative composition of the two network components [146]. The pH- and temperature-responsive hydrogel, pAsp-*l*-pNIPAm were also reported by Nemethy et al. for colon-targeted controlled drug delivery. This multi-responsive co-network hydrogel was synthesized by reacting allylamine-grafted polysuccinimide and NIPAAm in organic medium followed by hydrolysis [147]. Temperature- and pH-responses were investigated by swelling degree measurements. Scanning electron microscopic examination revealed that the morphology and pore size of the hydrogel could be regulated by the ambient temperature and pH. A commonly using non-steroid anti-inflammatory drug, diclofenac sodium (DFS) was loaded for the drug release analysis. The hydrogel exhibited high compatibility towards human epithelial cells and considered as appropriate tool for colon-targeted controlled delivery of drugs.

A pH-responsive semi-interpenetrating polymer network hydrogel was also reported for in vitro drug delivery [148]. The hydrogel was prepared by using konjac glucomannan and pAsp with trisodium trimetaphosphate as a crosslinking agent. At pH 2.2, the release of 5-fluoruracil was about 23% and approached to 95% at pH 7.4. Citraconylated pAsp was utilized to develop a smart drug delivery system displaying a dual pH-responsive property. Hydrazone linker and citraconic amide effectively assisted the release of drug at low pH [149]. pAsp was also utilized for ophthalmic drug delivery application. An in vitro test was conducted to study the ophthalmic formulation aspects using a thiolated pAsp. The results confirmed the potential of these polymers as carriers for ophthalmic drug delivery [150]. pAsp sequence-linked biodegradable polymeric NPs can interact precisely with mineralized tissues. Moreover, a single Asp based-osteotropic nanoscale drug delivery system reported for the bone-targeting purpose [151]. Polyaspartylhydrazide copolymer were used as folate-targeted supramolecular vesicular aggregates for antitumoral drug delivery. Gemcitabine- loaded vesicular aggregates showed significant anticancer activity and were removed from the circulatory system at a slower rate than the native drug. The vesicular aggregates were remarkably delivered to kidney, lungs, spleen and brain [152].

Hydrogels based on inulin and α,β-polyaspartylhydrazide (pAspHz) were used for colonic drug delivery. Succinic derivatives of inulin (INU-SA) with two different degrees of functionality were cross-linked with pAapHz to obtain INU-pAspHz hydrogels and then GSH and oxytocin, for the oral treatment of inflammatory bowel disease, were loaded [153]. The incorporation of pAsp to the protein-based drug delivery platform apoferritin created the negative charge at pH 5.0, enhancing the drug encapsulation. This system had a potential as therapeutic administration of the anti-cancer agent daunomycin [154]. Stimuli-responsive hydrogels were synthesized using pAsp and used for colon-specific drug delivery [155]. 5-Flurouracil were loaded in the hydrogel exhibited excellent pH-responsive drug releases, suitable platform for oral treatment of colonic tumor. Wang et al. reported a pH-responsive zwitterionic pAA derivative for drug delivery. It was prepared by a partial conjugation of *N*-(3-aminopropyl)morpholine, methyl His, *N*-(3-aminopropyl)dimethylamine and ethylenediamine to the side chains of pAsp [156]. An interesting polymer micelle system was developed to demonstrate the encapsulation and stabilization of wide variety of hydrophobic drugs [157]. pAsp middle block reacted with Fe to form metal acetate bonds at high pH. This system were capable to dissociate easily at low pH (Figure 7).

Biodegradable polymeric NPs were fabricated using pAsp and poly(lactic acid) conjugated with 1,2-dialmitoyl-sn-glycero-3-phosphoethanolamine (DPPE) to encapsulate Dox. These NPs displayed excellent intracellular delivery in tumor cells with co-localization in lysosome and showed delayed nucleus entry depending on the pH [158]. An advanced micellar drug delivery platform with better sustained release and compatibility was fabricated with a pAsp-based amphiphilic copolymer. Dox and mPEG were conjugated onto pAspHz to afford an amphiphilic polymer containing acid-cleavable hydrazone bonds. The designing of Dox conjugated pAspHz is in order to supply hydrophobic segments. While hydrophilic segments were supplied by grafting of mPEGs to the polymer via hydrazone bonds and there by prolonged circulation time in blood. Dox loading capacity was raised to 38% by the feed ratio controlling. The in vitro profiles confirmed a much faster drug release at pH 5.0 than that at pH 7.4. Moreover, the Dox-loaded system facilitates the long-time antitumor activity than free Dox.

pAsp derivatives improved incompatibility and sustained release of Dox. An amphiphilic polymer was synthesized by mPEG and Dox were conjugated onto pAsp. The drug delivery system exhibit biodegradability, biocompatibility, high drug loading efficacy and pH-responsiveness [159]. The pAsp derivatives were used for synthesizing a pH-dependent and charge-converting nano-carriers by conjugating 2,3-dimethylmaleic anhydride to the amine group of an octadecyl-grafted poly(2-hydoxyethyl aspartamide) backbone. Dox successfully encapsulated in the spherical micelles was subjected to evaluate tumor cell uptake and anticancer therapeutic efficiency [160]. pAsp-based polyanionic nano-carriers were reported for oral administration. PSI was firstly synthesized from pAsp using acid-catalyzed bulk thermal polycondensation in a mixture of mesitylene/sulfonate. The resultant polyanions self-assembled to stable nano-scale micelles with tunable size depending on pH [161]. A cancer targeting triggered drug delivery to tumor cells was demonstrated with an innovative mesoporous silica nanoparticle (MEMSN) envelope-type cellular-uptake-shielding multifunctional system [162]. Mesoporous NP surfaces was anchored with β-cyclodextrin via disulfide linking for GSH-induced intracellular drug release. Then via host–guest interaction, the surface of the NP was again altered with a RGD peptide sequence and matrix metalloproteinase substrate peptide Pro-Leu-Gly-Val-Arg (PLGVR). The NP were again modified with pAsp to obtain MEMSN, which have the ability of protecting the targeting ligand. In addition, MEMSN prevent the uptake by normal cells, which was confirmed by in vitro study. After reaching the tumor cells, pAsp protection layer was removed via hydrolyzation of PLGVR at the metalloproteinase-rich cancer cells. Thus, the targeting property could be switched on, which permitted the easy uptake of drug-encapsulated NP by cancer cells followed by GSH-induced intracellular release of drug (Figure 8).

PEG-*b*-pAsp copolymer was also used as a shielding agent for DTX-loaded NPs to deliver DTX to drug resistant cancer cells. DTX was encapsulated into the lipid core of the 200 nm sized NP of unique lipid polymer hybrid, which subsequently protected with the pH-responsive block copolymer PEG-*b*-pAsp [163]. Drugs were released from DTX-loaded NP in a pH-sensitive way and the NPs were capable to efficiently induce cytotoxicity in tumor cells. The blood circulation and physiological activity of DTX incorporated PEG-*b*-pAsp NPs were strikingly enhanced by the surface negative charge and PEG shell of NPs. The high concentration of caspase-3 and poly(ADP-ribose) polymerase-1 detected in the tumor cell after treatment confirmed the in vivo anticancer efficacy. Thus, this novel lipid polymer hybrid system could be an efficient platform for cancer treatment.

A superabsorbent hydrogel was developed from the pAsp using chemical crosslinking with diamine as a crosslinking agent. Water absorbance was compared at pH 8 and 10, demonstrating that the absorbent capacity increased with increasing pH and PSI concentration. Furthermore, the hydrogel showed ampholytic and reversible pH-response properties [164]. pAsp modified with cystamine was developed as a pH and redox-responsive hydrogel by Gyarmati et al. [165] This hydrogel achieved a reversible gelation and dissolution in dimethylformamide and in aqueous medium through a thiol-disulfide interconversion of the side chain of polymers. It converted to a mechanically stable gel by oxidation, which are suitable for drug incorporation and site specific drug delivery.

Hydrogels showing a reversible response to external pH and redox stimuli were also developed by using pAsp moiety. The reversible responses of the degree of swelling and the elastic modulus were triggered through the thiol-disulfide transformation inside the hydrogel [166]. A novel pH-responsive proteasome inhibitor delivery carrier was developed using PEG and pAsp copolymer with acid labile hydrazone bond. The proteasome inhibitor MG132 covalently bound to the matrix showed a remarkable antitumor effect [167].

A hydrazine-cross-linked pH-responsive zwitterionic nanogel was designed for controlled drug delivery. Outstanding anti-protein adsorption capability and stability in protein solution were characteristics of this zwitterionic pAsp naonogel (Figure 9). The final nanogel was emphasized by pH-response, anti-protein adsorption ability and biocompatibility [168]. The pAsp derived PSI were used to develop pH-responsive NP for site-specific delivery in agriculture. PSI was functionalized with primary amines (2-(2-aminoethoxy)ethanol, hexylamine) to form random amphiphilic copolymers [169]. Very recently, our group published a GSH- and endosomal pH-responsive hybrid vesicle for anticancer delivery platform using pAsp [170]. A zwitterionic/amphiphilic block copolymer, poly(2-methacryloyloxyethyl phosphorylcholine)_50_-*b*-pAsp*_n_* containing a degradable disulfide linker was synthesized and investigated its dual-stimuli-responsive Dox delivery. Dox was effectively loaded into 100 nm sized uniform vesicles and the release nature was examined under various pH conditions and in different concentrations of the reducing agent, 1,4-dithiothreitol (DTT). At physiological conditions, faster Dox release was obtained with increasing concentrations of DTT and Dox release was maximum in high DTT concentrations at pH 5.5 (Figure 10).

The combined delivery of drug and gene was attempted using pAsp. Electrostatic attraction between protamine sulfate and pAsp or Dox-conjugated pAsp led to self-assemble to form regular spherical shapes. When PS/pAsp > 2, DNA was completely bound to PICs and quickly released into the HeLa cells [171]. A protein drug delivery system was possible using pAsp and quarternized chitosan [172]. Ionotropic gelation was utilized to fabricate NPs. BSA was used as a model protein for entrapment studies. The release was fast at pH 7.4; however, became slow at pH 1.2.

Hakeem et al. demonstrated pAsp-anchored mesoporous silica NPs as a pH-responsive drug delivery system [173]. Self-propelled and targeted drug delivery of pAsp/iron-zinc micro-rocket in the stomach was also reported [174]. A microdevice consisting of a pAsp micro-tube, a thin Fe intermediate layer, and a core of zinc could be propelled consuming gastric acid as a fuel. Following the adsorbing Dox on a pAsp surface, the micro-rocket carried drugs, magnetically detect targets, infused the gastric mucus gel layer, and enhanced drug retention in the stomach without inducing an obvious toxic reaction. Micelles of 15–30 nm in size fabricated by pAsp functionalized with primary amine, arginine and fatty amine entered into the cell by lipid raft endocytosis, moved to perinuclear region where polyglutamine aggregate or amyloid oligomer predominantly localize, remove the aggregated protein from the cell and improves the cell survival against toxic polyglutamine/amyloid aggregates [175]. This system has a potential as a promising drug delivery carrier for improved efficiency in the treatment of neurodegenerative diseases using anti-amyloidogenic drugs.

#### 2.2.2. Poly(glutamic acid)-Based Polypeptides

Glutamic acid (Glu) is an essential α-AA in humans and the same is used by most of the living creatures in protein biosynthesis. Furthermore, it is the most abundant one in the vertebrate nervous system and acts an excitatory neurotransmitter. In the moderate acidic water solutions and solid state, the molecule adopts an electrically neutral zwitterion structure. From the second carboxyl group, the acid can lose one proton to form the conjugate base, the singly-negative anion glutamate. This compound form is prevalent in neutral solutions. The glutamate neurotransmitter demonstrate the major role in neural activation. Poly(l-glutamic acid) (pGlu) facilitates antiparallel association characteristics of helical rods, which are useful for the incorporation of functionalites like DNA-intercalatable and redox active groups at the chain end [176,177,178].

The manufacture of pH-responsive fullerene-containing pGlu NPs were described to investigate their superoxide dismutase mimetic property (Figure 11) [179]. The fullerene peptide self-assembled to form NPs of 100–200 nm in diameter in pH < 6.8. These NPs were rich in stacking interaction of fullerene moieties and alpha-helices, contributed to the stability of the high-order structure. The fullerene-containing NP was capable to eliminate the biologically important superoxide radical in contrast with the superoxide dismutase.

Double hydrophilic and dual responsive cross-linked micelles were fabricated using PLGA as a drug delivery vehicle [180]. The graft copolymer was firstly prepared by sequential grafting of pNIPAm and 2-hydroxyethyl methacrylate onto PLGA backbone. The protonation of PLGA at pH 5.0 induced the enhanced hydrophobic interactions, leading to aggregation of micelles. Phase transition was induced due to the temperature responsive characteristics of pNIPAm. Considering PLGA has been established to be biodegradable both in vitro and in vivo, the crosslinked micelles have promising applications in intelligent drug delivery systems. Recently, Yang et al. reported a hydrogel made from PLGA and NIPAm in the presence of allyl glycidyl ether (AGE) as a crosslinker [181]. As the amount of PLGA-AGE increased, the cytocompatibility of hydrogel increased. Zhao et al. used a new strategy for developing the thermo- and pH-responsive hydrogels using the crosslinking of pNIPAm with a biodegradable crosslinker derived from PLGA. The hydrogels synthesized by exposing aqueous solutions of precursor containing photo-initiator to UV light irradiation shrank under acidic pH or at temperature higher than collapse temperature and would swell in neutral or basic media or at lower temperature. Interestingly the hydrogels indicated no weight loss in the simulated gastric fluid but disintegrated rapidly in the simulated state of intestine. The release of BSA loaded into the hydrogels was both pH- and temperature-dependent [182]. A triple responsive nanogels were also reported based on PLGA as a potential intracellular drug carrier. mPEG-*b*-poly(l-glutamic acid-*co*-γ-2-chloroethyl-l-glutamate) mPEG-*b*-p(LG-*co*-CELG)) micelle was prepared by crosslinking using the dual redox responsive diselenide bond [183]. Dox release studies and MTT assay exposed that the nanogel was biocompatible and the Dox release was enhanced by GSH.

A new class of PLGA-based star-block copolymers consisting of a hyperbranched polyethylene-imine core, a PLGA inner shell, and a PEG outer shell were manufactured as efficient drug carriers for effective incorporation of cationic payloads and for pH-responsive release [184]. The side chains of PLGA exist mostly in deprotonated form at physiological pH. The cationic adriamycin hydrochloride and crystal violet could be effectively incorporated via electrostatic interactions to the synthesized polymer. The encapsulated drugs in the star-block copolymer had a comparatively high temporal stability and displayed remarkably enhanced in low pH. A facile preparation of pH- and reduction-responsive polypeptide nanogel was suggested for an efficient drug loading and an intracellular delivery [185]. mPEG, PLGA and poly(*l*-cystine) (pCys) were used to prepare the nanogels. The nanogels were biocompatible and potent to utilize as effective drug carrier platform. Introduction of hydrophobic spacers to PLGA significantly enhanced its interaction with lectin and its helicity in acidic medium, leading typical coil-to-helix transition [186]. Dox was encapsulated into the NPs fabricated using an amphiphilic copolymer containing PLGA, 1,2-dipalmitoyl-*sn*-glycero-3-phosphoethanolamine, and polylactide segments by double emulsion and nanoprecipitation methods [187]. 

Dox-loaded NPs demonstrated a faster Dox release at pH 5.0 compared to pH 7.4. Dox-encapsulated NPs exhibited effectively time-delayed in vitro cytotoxicity against Hela and C666-1. In contrast to free Dox, the Dox-loaded NPs could selectively accumulate in lysosomes. Recently a star-shaped porphyrin-cored PLGA conjugates was developed as an efficient photosensitizer. It was synthesized via ROP of β-benzyl-l-glutamate NCA monomer with 5,10,15,20-tetrakis-(4-aminophenyl)-21H,23H porphyrin (TAPP) as an initiator. It was found that the fluorescence intensity of the nanocarrier increased with decreasing pH and the PLGA improved the solubility of porphyrin in water along with accelerated production of singlet oxygen [188].

A fluorescence resonance energy transfer (FRET)-based dual-emission and pH-responsive nano-carrier was reported for improved protein delivery to overcome the intestinal epithelial cell barrier (Figure 12). This nanocarrier, fabricated by combining chitosan-*N*-arginine and PLGA-taurine conjugates. exhibited a pH responsive fluorescent color and emission intensity variation [189]. PLGA used as gene-carriers using pH-responsive and zwitterionic properties. PEI-PLL-PLGA was used to design the shielding system for gene carriers. The smart polymer shielded PEI (MW=125,000)/DNA by forming ternary polyplex that aids electrostatic interactions between negatively charged cancer cells and positive polyplexes, leading to high cell uptake [62]. PLGA functionalization of chitosan NPs using poly(sodium 4-styrenesulfonate) as a cross-linking agent enhanced the therapeutic efficacy of insulin following oral administration [190]. The PLGA functionalization improved the oral uptake through calcium-sensing receptors and AA transporters present in intestinal epithelium. At the optimized formulation using the design of experiments (DoE) approach, spherical NPs with particle size, zeta potential, and entrapment efficiency in the range of 210 nm ± 2.8 nm, 18.1 mV ± 0.14 mV, and 85.9% ± 0.28%, were formed, respectively. In vivo studies in diabetic animals demonstrated low levels of plasma glucose for almost 24 h. Pharmacological availability of insulin administered through the NPs was considerably higher than that of insulin administered through control NPs, overcoming the poor stability and, thus, poor therapeutic efficacy following oral administration.

Recently an injectable formulation consisting of a drug-loaded polymeric nanocapsules and low MW gelator-based hydrogel was reported [191]. The nanocapsules fabricated by using HA and PLGA and encapsulated with ^14^C-gemcitabine exhibited a size of 40 and 80 nm with more than 90% loading efficiency. These NCs shown significant control on the release of the encapsulated drug in such a way that the release was continued for more than one month. Moreover, 4-*N*-myristoyl-^14^C-gemcitabine encapsulated NCs exhibited excellent activity against different cancer cell lines. This nanocomposite formulation exhibited unique characteristics such as capacity to regulate the release of the nanocapsules, adequate mechanical strength, and gel formation ability in situ upon injection. The nanocomposite formulation into a gel transformation was occurred instantaneously after injection. Thus, this nanocapsules is a promising tool for intracancer delivery of antitumor drugs in unresectable tumors such as oesophageal, pancreatic or gastric tumor. Different micelles formed by the self-assembly of anionic polypeptide- based polymers are summarized in Table 3.

Recently, the incidence of thrombotic diseases slowly growing, becoming a serious threat to the human health since thrombosis has more chance to cause cardiovascular diseases. In order to use for protecting and transporting nattokinase (NK) that is a thrombolytic drug with efficient thrombolytic effect with less side effects, mesoporous silica/PLGA peptide dendrimer with dual targeting was designed [192]. Magnetic mesoporous silica NPs were firstly prepared which acts as the core, and PLGA peptide dendrimer was conjugated. RGD was then grafted onto it. Thrombus-targeting nanocomposites were fabricated by encapsulating the drug NK via electrostatic interaction. In vitro and in vivo targeted thrombolytic studies confirmed the noteworthy thrombolysis capability of the NPs. Various anionic polypeptide-based pH-responsive drug release systems are summarized in Table 4.

## 3. pH-Responsive Drug Release in the Organ Level

Among the various methods for the oral drug administration, the pH-responsive drug delivery is of prime importance because of the variety of pH in the gastrointestinal tract. Thus, it is possible to release the drug in targeted organs using the suitable pH-responsive carriers. The pH-responsive oral drug delivery techniques have been noticeably improved past decades. The importance of organ level drug delivery is due to the convenience, cost-effectiveness and better compliance from patients, particularly for chronic diseases that necessitate frequent drug administration, even though hydrolysis and enzymatic degradation in the gastrointestinal tract become a bottleneck for this application [193]. The pH-responsive carriers keeping their stability in these circumstances have potential to use for oral drug delivery such as vaccination and inflammatory bowel diseases. In some cases, compared to NPs, multi-ion-cross linked NPs more effectively transport insulin-like drug [194].

Polyanionic peptides such as PLGA and pAsp are specially interesting for fabricating NPs for oral drug delivery. The NPs formed in a certain pH range tend to disintegrate at lower pH due to the protonation of ionized carboxyl group. They aid to provide the moderate process to obstruct drug denaturation and to enhance the specific site absorption. Zheng et al. reported NPs based on the complex of chitosan and pAsp sodium salt [195]. The size (85–300 nm) and morphological structure of the polyelectrolyte chitosan-PAsp NPs was controllable by varying chitosan to pAsp ratio, cross-linker concentration, ionic strength, incubation temperature, and solution pH. After the loading of a hydrophilic drug, 5-fluorouracil (5FU), in vitro and in vivo experiments of the drug-loaded NPs showed a sustained release of 5FU.

The pH-responsive NPs obtained by self-assembly of poly(γ-glutamic acid) (γ-PGA) chitosan and were used for the oral delivery of a monomeric form of insulin analogue (Asp-insulin) to enhance the intestinal paracellular transport [196]. The biodistribution of Asp-insulin for the oral administration in a rat model was studied using the single-photon emission computed tomography/computed tomography. Asp-insulin was absorbed into the systemic circulation when administered through oral, while chitosan was non-adsorbed remain in the gastrointestinal tract. After 20 min of injection via subcutaneous administration, the maximum Asp-insulin concentration in the peripheral tissue/plasma was noticed (Figure 13). Half of the initial dose was noticed to be degraded and excreted within 3 h. During the course of study, orally administration exhibited constant circulation of Asp-insulin in the peripheral tissue/plasma, while 20% of the initial dose of Asp-insulin was noticed as metabolized and excreted. The pharmacokinetic evaluation and pharmacodynamics of the Asp-insulin by oral administration with those of the subcutaneous injection of NPH-insulin (or isophane insulin) that is an intermediate–acting insulin given to assist blood sugar level controlling in people with diabetes suggested the significance of this NP system to be used as a non-invasive alternative for the basal insulin therapy.

Common antibiotics lke tetracycline, metronidazole and amoxicillin are worthwhile in treating *Helicobacter pylori* (*H. pylori*) in vitro; however, due to the fast degradation of the antibiotic in the acidic gastric environment and the associated trouble of attaining minimum inhibitory concentrations for *H. pylori*, they are frequently ineffective when used to treat infections in vivo. As a means of overcoming the problems, Chang et al. designed chitosan/γ-PGA NPs encapsulated into pH-sensitive hydrogels as an excellent platform for amoxicillin delivery [197]. The pH-responsive hydrogels protected NPs from being destroyed by gastric acid. In vitro drug release tests showed the quantity of amoxicillin released from NPs incorporated in hydrogels was relatively low (14%) at pH 1.2, compared to that from only NPs (50%), because NPs could penetrate cell–cell intersections and interact with *H. pylori* infection sites in the intercellular spaces and amoxicillin-loaded NPs in a hydrogel shiled the drug from the enzymatic degradation and assisted amoxicillin interaction specifically in the site of *H. pylori* infection. For the effective oral administration of antitumor drugs, nanocarriers need to endure in stomach, firmly hold the drug in the bloodstream, effeciently penetrate across the small intestine and rapidly release the drug in cancer cells. In order to fullfil these requirements a approach for oral chemotherapy was proposed using dual pH-sensitive polyelectrolyte complex NPs that attain intestinal penetrability, gastric survivability, intracellular activity and stability in the bloodstream [198]. The electrostatic interaction between negative PLGA grafted PEG-Dox conjugate and positively charged chitosan forms targeting NPs with acid-labile hydrazone linkages. The resultant NPs with the protection of chitosan layer assisted to survive in the stomach and pH-induced deprotonation helps to infiltrate through the small intestine into blood circulation after detaching chitosan chains from the complex NPs. In addition the NPs had long-circulating properties so that they could be effectively and rapidly accumulated in tumor tissue due to intracellular acid-trigged hydrolysis of hydrazone bonds, accomplishing a much better cancer therapy efficacy with less side effects compared to free Dox.

## 4. pH-Responsive Drug Release in the Tissue Level

Tumor tissues have an acidic pH range from 5.7 to 7.8 is due to the lactic acid accumulation in quickly growing cancer cells due to their high rate of glucose intake and low rate of oxidative phosphorylation and insufficient blood supply along with poor lymphatic drainage. These characteristic features are utilized for targeted drug release using nanocarriers which are having pH-responsive physical and chemical property-changing ability. One of the main working mechanisms is hydrophobic to hydrophilic transitions allowing polymer dissolution and subsequent polymer matrix collapse for drug release. Therefore, the demand for tumor-responsive nanocarriers, especially pH-responsive smart nanocarriers is increasing. PLGA is the main peptide that can effectively accumulate the payloads in tumor tissues. Wang et al. reported a PLGA-*b*-poly(d,l-lactic acid) to form Dox loaded hybrid micelles to treat melanoma by intracellular drug delivery (Figure 14). Under tumor-acidic conditions PLGA block undergoes conformational changes to trigger the Dox release [199].

Another pAA-based amphiphilic copolymer with pH and thermo-sensitive properties was loaded with Dox exhibited increased release under acidic conditions. The delivery system was composed of polyaspartamide scaffold functionalized with *N,N*-diisopropylamide groups by aminolysis of PSI and the pAA-shielded with PEG chains with acid-labile hydrazone linkages [200]. A site-specific drug-releasing polypeptide platform (PEG-pHis/pAsp-Dox/CA4) was developed by Dong et al. to improve the drug accumulation in drug resistant cancer cells, and the platform exhibited a dual-pH-response and remarkable cytotoxicity against Dox-resistant cells [201].

## 5. pH-Responsive Drug Release at the Cellular Level

At the cellular level, endosomal acidification results in decreasing cellular pH to less than 6.0 owing to a vacuolar proton ATPase-assisted proton influx [202]. The NPs show a capability of intracellular delivery, because they get protonated by a proton sponge effect at endosomal pH causing an osmotic pressure hike in the endosomal compartment. Finally, the NPs are released in cytoplasm after the disruption of the plasma membrane [203]. Thus, biocompatible polypeptide NPs having pH-responsive features with endosomal acidification are considered as efficient platforms for enhanced drug release to the cytoplasm.

To achieve the effective intracellular delivery of Dox, Chiang et al. prepared a hollow lipid/polypeptide conjugate with a dual-layered nanogel-coating. The gel-caged polymeric vesicles from γ-PGA exhibit cytotoxicity in cellular level by quick drug release in acidic lysosomes and endosomes [204]. The endosomal pH-responsive intracellular release of Paclitaxel^®^ was demonstrated by Gao et al. using a dual-ligand-modified micelle with cyclic Arg-Gly-Asp-Tyr-Lys. This formed 30 nm micelles and showed rapid intracellular drug release at endo/lysosome pH [205]. The alternation of ionization state of the carboxylic groups in the AA is considered to be the quick release of payloads in the endo/lysosome pH. Pepsomes (polypeptide-based polymersomes) fabricated by PEG-*b*-poly(l-leucine)-*b*-PLGA presented fast disruption and Dox release due to the shift of hydrophilic nature and random coil structure of PLGA into hydrophobic and α-helical structures. Confocal microscopy confirmed the Dox release into the cell nuclei of MCF-7/ADR cells in 4 h [206]. Another acidic pH-triggered liposomal delivery was reported using a newly designed hydrophilic anti-human thymidylate synthase by reducing the multi-drug resistance in cisplatin resistant cancer cells [207]. pAsp based gene delivery vehicle conjugated with octa-Arg peptide with pH and redox dual-responsive characteristics was recently designed. The octa-Arg peptide facilitated the intracellular nucleus import of DNA, confirming the significance of intelligent vector in the transfection of non-dividing cells [208]. The positive charges of most of the cell penetrating peptides usually lead to nonspecific binding and high toxicity. Recently, an anionic, Lys-based hyperbranched polymer mimicking peptides are developed for overcoming the barrier in the case of delivery of drugs to cytoplasm. These novel carriers altered conformation according to pH together with its multivalence effect to facilitate the complete membrane disruption at late endosomal pH (Figure 15) [209].

## 6. Combination with Other Responsive Materials

Multi-stimuli-responsive peptides can respond to either two or more stimuli such as pH, temperature, enzyme, light, or redox. Thus, it is possible to develop novel nanocarriers with well-tuned responsive properties that efficiently release their payloads at specific sites utilizing this multi-stimuli-responsiveness. Incorporation of unprecedented levels of functionality into polypeptides facilitates the development of multi-responsive nanocarriers. Among these, redox responsiveness recently gets much attention especially for anti-tumor drug delivery. The presence of high GSH content in cancer cells was effectively involved in the cleavage of redox responsive linkages and faster release of drugs [210]. Mainly disulfide linkages are utilized for redox responsive cleavage, while hydrazone linkages are susceptible to cleavage in acidic environment.

Tumor targeting nanocarriers are highly demanded for smart delivery, especially in the field of photodynamic therapy. A two-dimensional MoS_2_-based nanosystem was used for tumor therapy (Figure 16) by modifying with pH-responsive charge-convertible peptide and photosensitizer toluidine blue [211]. Photosensitizers like Ce6, protoporhyrin IX and verteporfin were encapsulated in nanospheres for pH, redox and proteinase concentration response capability, respectively. The multi-triggered release of photosensitizer in tumor tissues is favored for the photodynamic tumor ablation [212]. pH and photo-responsive polypeptides are promising platforms for overcoming the problems associated with cationic peptides and their poor cell selectivity. A photocleavable group is utilized to link a cationic peptide and an anionic peptide to enhance internalization into tumor cells by NIR light illumination and low pH at the tumor site [213].

Another important way to prepare smart carriers for drug delivery is the use of enzyme- mediated cleavage [214]. The site-specific controlled drug delivery can be improved by using enzymatically digestible hydrogel along with polypeptides composed of l-Glu and l-Lys. Hydrogel loaded with Dox and diclofenac sodium showed accelerated release of drug in the presence of trypsin, acidic pH and ionic strength [215]. The enzyme-degradability of the polymer could be controlled by pH induced structural conversion from polyester to polypeptide [216].

## 7. Conclusions and Outlook

In nature, proteins are some of the most important biomaterials and play various cell functions such as catalysis, signaling and transportation of molecules by adapting higher order structures. This has inspired researchers towards synthetic molecules derived from the biomolecules to mimic proteins. Synthetic polypeptides are the most widely recognized protein mimicking materials for biomedical applications. In this review article, we have investigated the recent advances in the design and synthesis of polypeptide nano-carriers derived from the AAs. The pH-responsive polypeptide-based nanocarriers were specifically discussed in detail. The number of recent publications on the smart pH-responsive polypeptides is the proof for the significance of this area of research. We have discussed the advantages of pH-responsive polypeptides, which are regarded as the main inducement for the rapidly growing drug delivery application. The extensive use of polypeptides to build variety of complex molecular architectures for biotechnological applications have been explored. Various applications of pH-responsive polypeptides in drug delivery at the organ, tissue and cell level are explored. This is mostly due to their biocompatibility, biodegradability and stimuli responsiveness. Furthermore, the incorporation of the various functional groups on the side chains of the AA residues simplifies the modification of polymer structures and helps to improve the efficiency of the nanocarrier in theranostic applications. In addition, polypeptides play key roles in polymer self-assembly to nanostructures like spherical micelles, vesicles, nanofibers, hydrogel, and nanotubes with significant drug loading efficacy.

Despite the abundance of studies on stimuli-responsive drug delivery systems, challenges persist, requiring significant ingenuity to obtain the maximum results in biological applications. Effective drug loading and drug release are still not achieved by utilizing advanced polypeptide nanocarriers. New research is required to focus on enhancing the drug loading capacity of nanocarriers by continuing to search for new smart nanocarriers with maximum drug release using new environmental stimuli. Even though polypeptides improve the biocompatibility of nano-carriers, modification of polymers with organic and inorganic materials may increase the chance of toxicity during in vivo biostudies and thereby diminish the efficacy of the research when it reaches the application stage. The current synthetic protocols for the synthesis of sequence controlled and monodisperse polypeptides are challenging. For example, moisture-sensitive NCA monomers require stringent anhydrous conditions for polymerization and the formation of unwanted side products cannot be avoided. In addition, more effective and environmentally friendly protocols are to be invented for the economic synthesis of NCA monomers.

The design and fabrication of nanocarriers utilizing more biomaterials along with pH-responsive polypeptides, especially for contrasting agents and other stimuli-responsive functional groups require tremendous effort to achieve complete biosafety. Another significant issue is the complexity of nanocarrier preparation and the cost of production. To achieve the successful application of a medicine, the medicine should be available easily at a reasonable price. For this, effective studies should be conducted to accomplish the fabrication of polymer nanostructures by utilizing simple and inexpensive materials and methods. Challenges including stability and repeatability during mass production of pH-sensing systems also need to be seriously considered for better reliability and better efficacy. In addition to that the development of pH sensing systems with high sensitivity and quick response time are promising in the areas of diagnosis and treatment. The enhanced research in the academic and pharmaceutical area is expected to provide potential solutions for these challenges. Polypeptides can be modified with a variety of functionalities to mimic the properties of various molecules and to achieve tremendous characteristic features. Substantial reports on the synthesis of side-chain functional polymers have been noticed to be promising for the polypeptide-based nanocarrier platforms. Research activities in the development of new conjugation reactions are also expected to provide suitable side-functionalization for polypeptides to mimic natural biopolymers and show enhanced therapeutic efficiency.

## Figures and Tables

**Figure 1 molecules-24-02961-f001:**
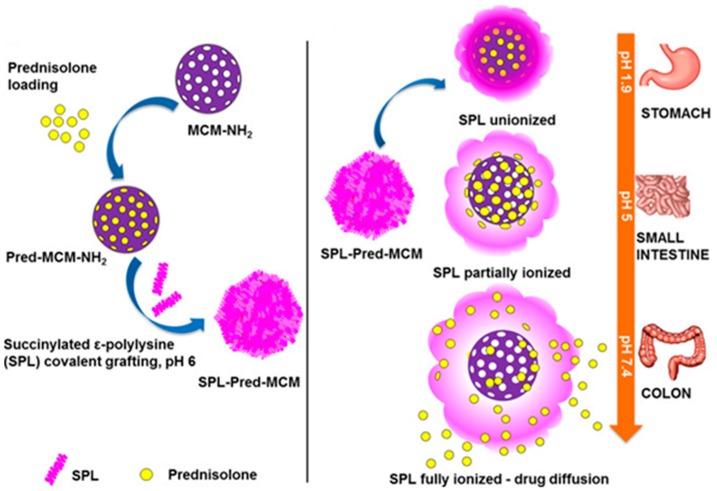
Schematic representation of the coating of succinylated ε-polylysine on to 3-aminopropyl-functionalized mesoporous silica NP loaded with prednisolone and its selective release in the colon. Reproduced with permission from [46]. Copyright 2017, American Chemical Society.

**Figure 2 molecules-24-02961-f002:**
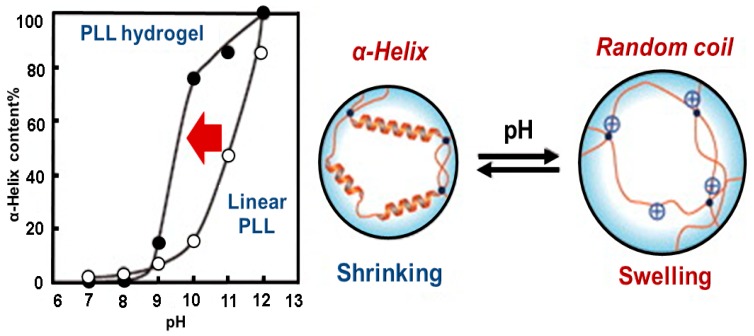
Illustration of the structural change of poly(l-lysine) from α-helix to random coil in response to pH change. Reproduced with permission from [54]. Copyright 2015, The Chemical Society of Japan.

**Figure 3 molecules-24-02961-f003:**
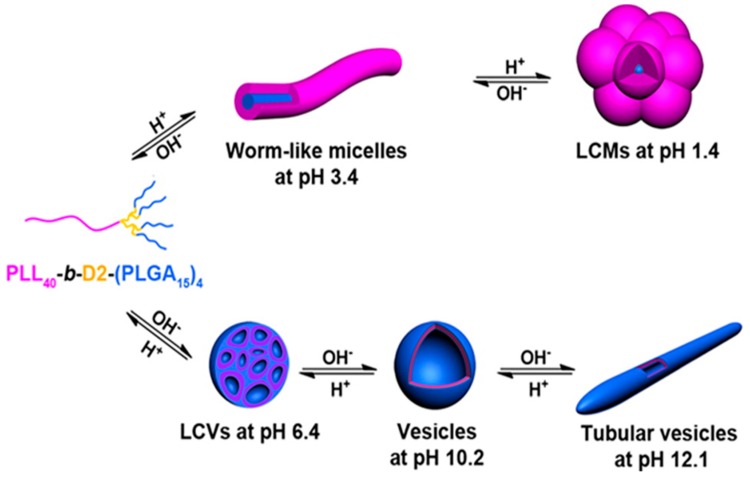
Schematic representation of pH-responsive self-assembly of linear-dendron-like polyampholyte, PLL-*b*-D_2_-(PLGA)_4_. Reproduced with permission from [55]. Copyright 2013, American Chemical Society.

**Figure 4 molecules-24-02961-f004:**
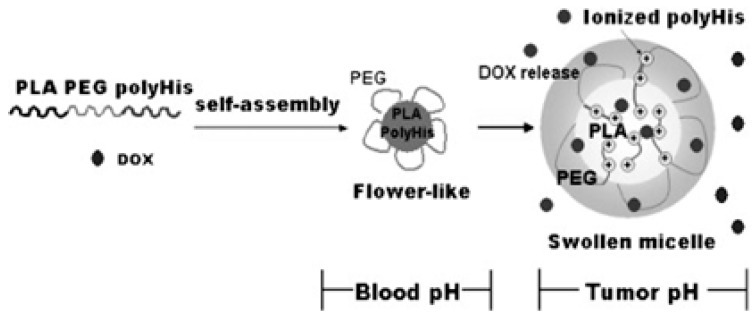
Schematic representation of formation of flower like micelle from poly(l-lactic acid)-*b*-poly(ethylene glycol)-*b*-poly(l-histidine) and triggered release of drugs in tumor pH by re-self-assembly. Reproduced with permission from [71]. Copyright 2007, Elsevier.

**Figure 5 molecules-24-02961-f005:**
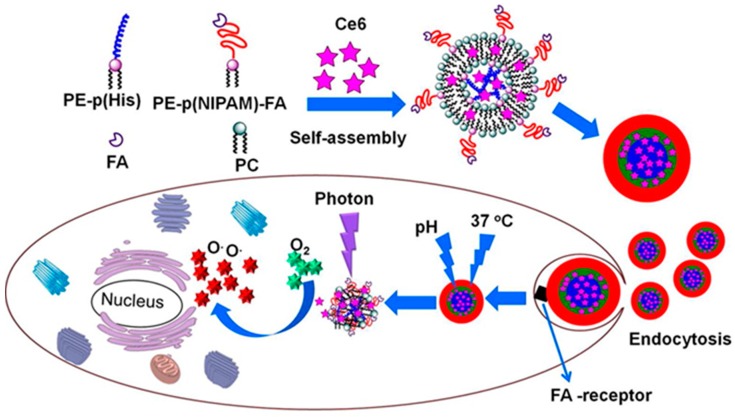
Schematic representation of photodynamic therapy utilizing the self-assemblies of the mixtures of lecithin derived phosphatidylcholine, phosphatidylethanolamine-poly(*l*-histidine)_40_ and folic acid conjugated phosphatidylethanolamine-poly(*N*-isopropylacrylamide)_40_. Reproduced with permission from [77]. Copyright 2016, American Chemical Society.

**Figure 6 molecules-24-02961-f006:**
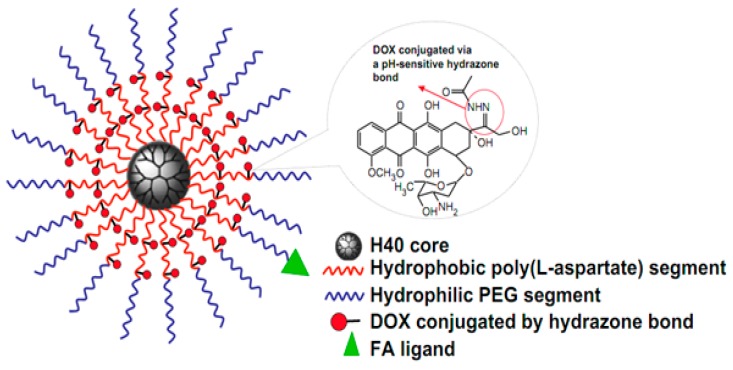
Diagram showing Dox conjugated multi-arm-block copolymer. Reproduced with permission from [141]. Copyright 2009, Elsevier.

**Figure 7 molecules-24-02961-f007:**
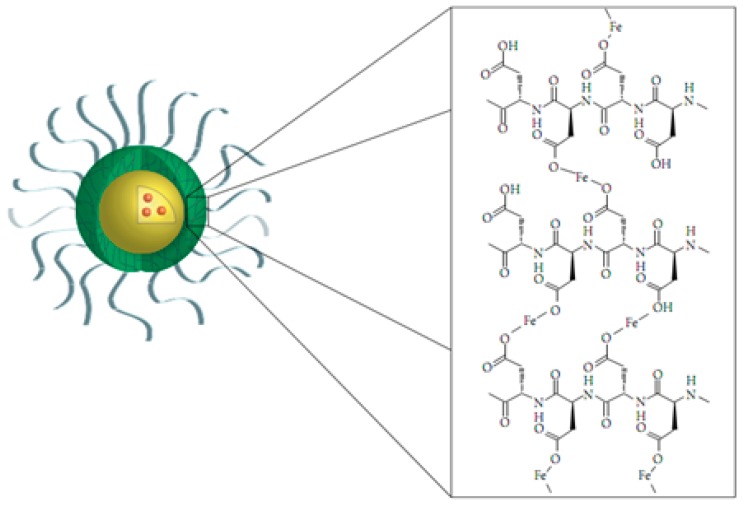
Diagram showing stabilized polymer micelles by cross linking the poly(aspartic acid) block with metal via metal acetate bonds. These bonds are break at low pH. Reproduced with permission from [157]. Copyright 2012 Jonathan Rios-Doria et al.

**Figure 8 molecules-24-02961-f008:**
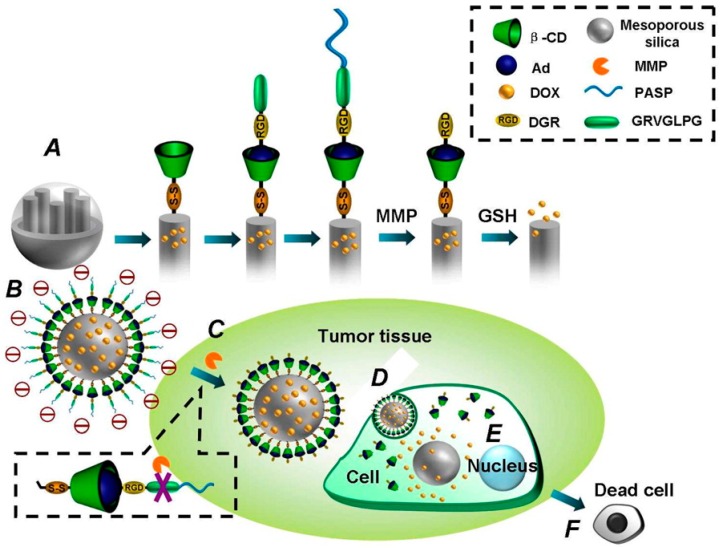
Structure of multifunctional envelope-type mesoporous silica nanoparticle and tumor-triggered targeting drug delivery. (**A**) Functionalization protocol of the mesoporous silica nanoparticle; (**B**) drugs loaded multifunctional envelope-type mesoporous silica nanoparticle under physiological condition; (**C**) removal of pAsp protection layer in response to matrix metalloproteinase at a tumor site; (**D**) cell uptake through a peptide sequence containing RGD-mediated interaction; (**E**) glutathione-triggered drug release inside the cell; and (**F**) apoptosis of tumor cells. Reproduced with permission from [162]. Copyright 2013, American Chemical Society.

**Figure 9 molecules-24-02961-f009:**
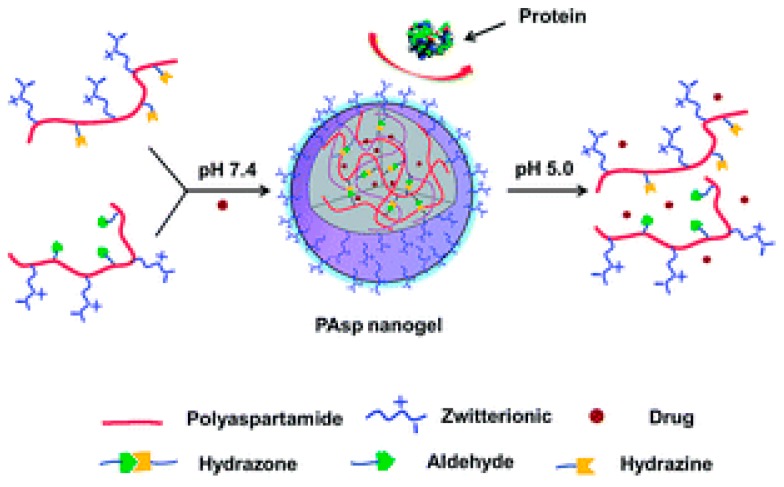
Schematic representation of the accelerated release of Dox from the zwitterionic polypeptide in acidic pH and its anti-adsorption against protein. Reproduced with permission from [168]. Copyright 2014, Royal Society of Chemistry.

**Figure 10 molecules-24-02961-f010:**
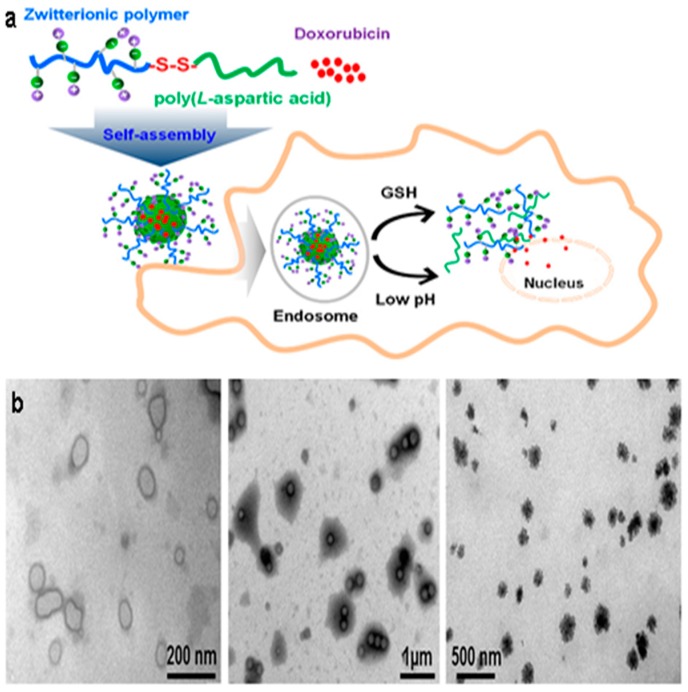
Schematic representation of the glutathione and endosomal pH-responsive release of drug from the poly(aspartic acid)-based hybrid vesicle (**a**) and TEM images of micelle at pH 7.4, 5.5 and at 5.5 in the presence of DTT (**b**). Reproduced with permission from [170]. Copyright 2017, Elsevier.

**Figure 11 molecules-24-02961-f011:**
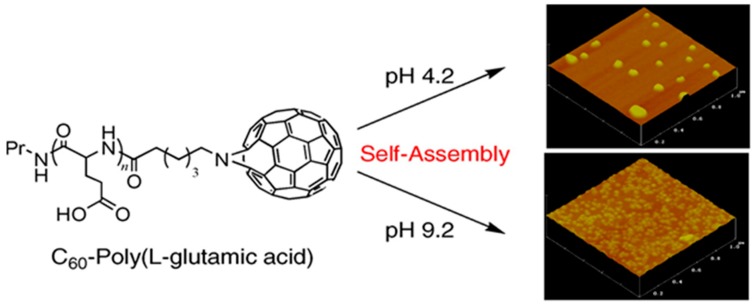
Illustration of fullerene-tagged poly(glutamic acid) and its pH-responsive self-assembling behavior. Reproduced with permission from [179]. Copyright 2006, Elsevier.

**Figure 12 molecules-24-02961-f012:**
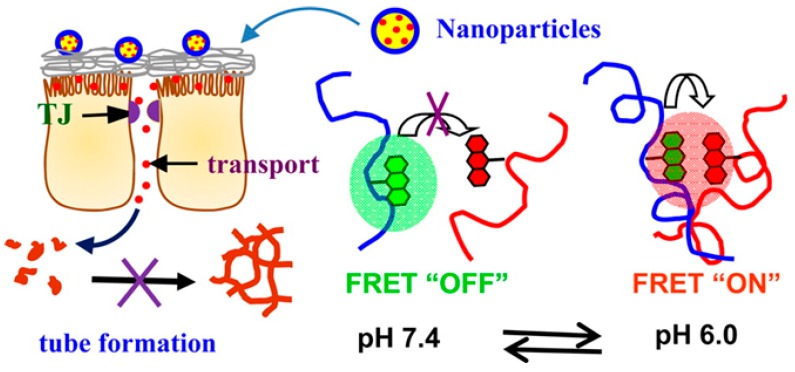
Schematic representation of dual emission and pH-responsive mediated drug release of FRET-based smart nanocarrier. Reproduced with permission from [189]. Copyright 2014, American Chemical Society.

**Figure 13 molecules-24-02961-f013:**
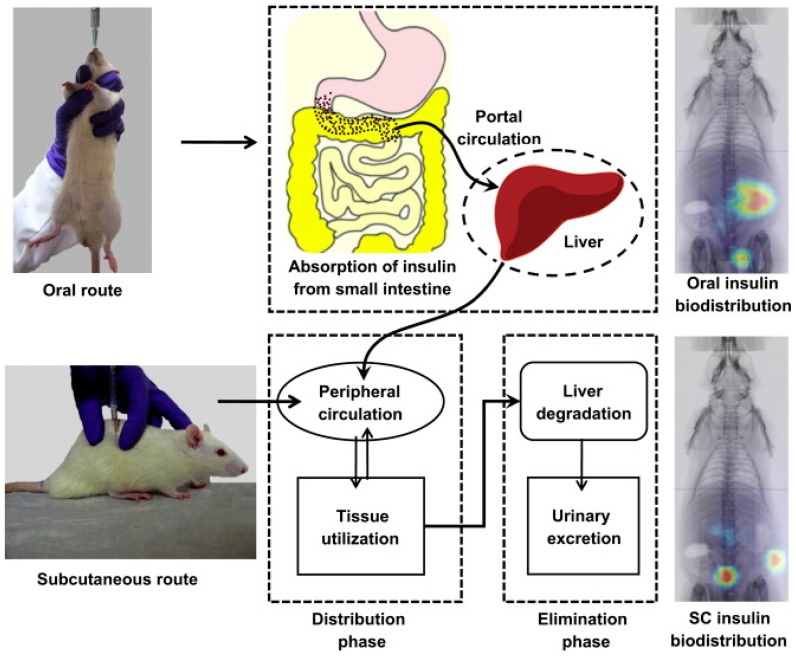
Schematic diagrams illustrating the absorption, distribution and elimination of aspart-insulin following oral or subcutaneous administration to rats. Reproduced with permission from [196]. Copyright 2010, Elsevier Ltd.

**Figure 14 molecules-24-02961-f014:**
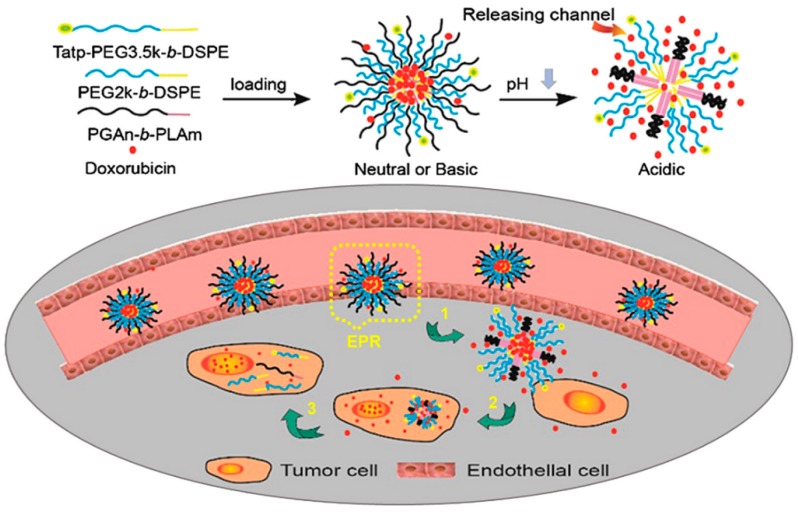
Schematic representation of intracellular delivery of doxorubicin using poly(l-glutamic acid) based hybrid micelles. Reproduced with permission from [199]. Copyright 2014, Elsevier Ltd.

**Figure 15 molecules-24-02961-f015:**
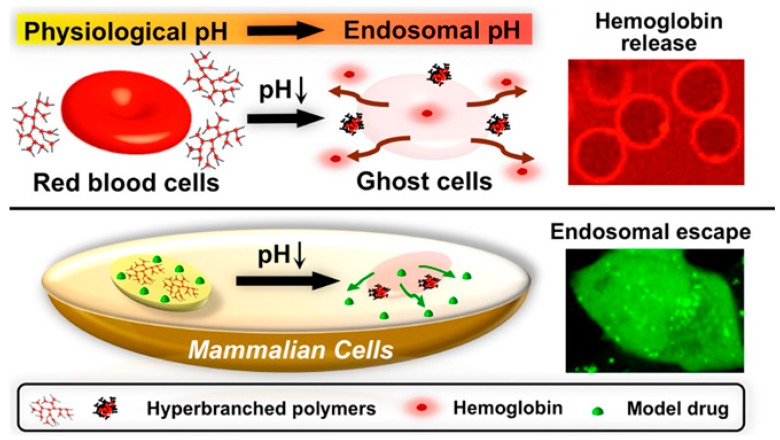
Illustration of the intracellular delivery of biological payloads from lysine-based hyperbranched polymers. Reproduced with permission from [209]. Copyright 2017, American Chemical Society.

**Figure 16 molecules-24-02961-f016:**
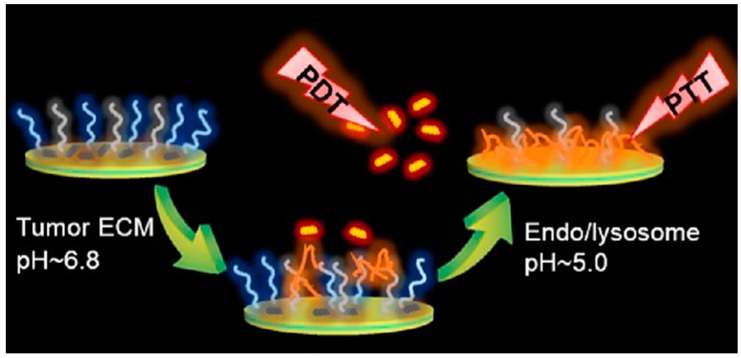
Illustration of the antitumor effect with toluidine blue O release and photo-induced reactive oxygen species generation of a multifunctional nanocarrier system in acidic pH. Reproduced with permission from [211]. Copyright 2017, American Chemical Society.

**Table 1 molecules-24-02961-t001:** Cationic polypeptide based self-assembled micelles.

Polymers ^a^	Micelle Types	Diameter	Ref.
PEG-*b*-PLL	Sphere	~65 nm	[47]
PPO-*b*-PLL	Vesicle	~80 nm	[49]
(PLL)_4_-d_2_-*b*-d_1_-(PLGA)_2_	Worm, vesicle, tubular vesicle, compound vesicle	16 nm–2 µm	[55]
PEI-*b*-PLL-*b*-PLGA	Polyplex	200 nm	[62]
pHis-*b*-PEI	Sphere	34 nm	[72]
Polysebacic anhydride-pHis	Nanocapsule	150–350 nm	[76]
p(MPC)-*b*-pHis	Nanodaisies	~100 nm	[79]
PEG-*b*-pHis-*b*-PLL	Rod	~325 × 13 nm	[81]
mPEG-*b*-PLA-pHis-S-S-OEI	Polyplex	116–658 nm	[88]
Alginate/pArg	Microcapsule	351 µm	[97]
HA-pArg	Sphere	116–155 nm	[103]
PArg	Nanocapsule	<200 nm	[104]
Lipid-pArg-PEG	Liposome	148–188 nm	[110]
mPEG-PLA-*b*-pArg	Micelleplex	54 nm	[111]
PEG-*b*-pTyr-lipoic acid	Sphere	45 nm	[124]
PEG-*b*-pTyr	Sphere	70 nm	[125]

^a^ See abbreviations for definitions of the terms used.

**Table 2 molecules-24-02961-t002:** Catonic amino acid derived polymers for pH-responsive drug release.

Type of AA	Payload	Target/pH	Polymer ^a^	Ref.
Lys	Prednisolone	Colon	Succinylated ε-PLL	[46]
	FITC-dextran	pH 3.5	Alginate bead coated with PLL	[50]
	Diclofenac sodium	pH 10-11 or 2-3	PEI-(PLL-*b*-PEG)	[51]
	Gene	pH 6.5	PEI-PLL-PLGA	[62]
His	Antitumor drug	pH 6.5	PLGA-*b*-PEG-*b*-pHis	[71]
	Dox	Tumor pH	Dextrin-*b*-pHis	[74]
	Ce6	Tumor pH	PE-pHis_40_ and PE- pNIPAm_40_-FA	[77]
Arg	Polyarginine-KLA peptide	Mitochondria	Polyarginine-KLA	[100]
	Docetaxel	Intracellular tumor	PArg nanocapsules	[83]
	Dox	Tumor pH	PAA-*g*-PEG/PArg	[106]
	Insulin	Intestine	pArg nano capsule	[107]
Tyr	Dox	Breast tumor	PEG-*b*-pTyr-lipoic acid	[124]
	Dox	Colorectal tumor	PEG-pTyr/cRGD-functionalized-PEG-pTyr	[125]
Pro	Nile blue A	pH 5.2	pNIPAm/pPro hydrogel	[128]
Trp	testosterone	pH 2.0	Poly(*γ*-benzyl-l-glutamate)-*graft*-poly(tryptophan)	[130]
	8-aniline-naphthalene-1-sulfonate & pyrene	pH 5.2	Poly(*N*-hydroxyethyl-l-glutamate)-*b*-poly(l-tryptophan)	[131]

^a^ See abbreviations for definitions of the terms used.

**Table 3 molecules-24-02961-t003:** Anionic polypeptide based self-assembled micelles.

Polymers ^a^	Micelle Types	Diameter	Ref.
Citraconylated-pAsp	Sphere	60 nm	[149]
FA-polyaspartylhydrazide	Vesicle	105–113 nm	[152]
pAsp-*co*-polylacticacid-DPPE	Sphere	219–281 nm	[158]
mPEG-pAsp	Vesicle	50 nm	[159]
PHEA-*g*-C_18_-NH_2_	Sphere	9 nm	[160]
PEG-*b*-pAsp	Sphere	200 nm	[163]
P(MPC)-*b*-pAsp	Vesicle	~100 nm	[170]
PLGA-pNIPAm-HEMA	Sphere	45 nm	[180]
PLGA-*co*-PLA-DPPE	Sphere	172–200 nm	[187]
Chitosan-*N*-arginine-PLGA	Sphere	260 nm	[189]
PLGA-chitosan	Sphere	210 nm	[190]

^a^ See abbreviations for definitions of the terms used.

**Table 4 molecules-24-02961-t004:** Anionic amino acid derived polymers for pH-responsive drug release.

Type of AA	Payload	Target/pH	Polymer ^a^	Ref.
Asp	DFS	Colon	pAsp-*l*-pNIPAm	[146]
	5-Fluoruracil	Intestine	Konjac glucomannan-pAsp hydrogel	[147]
	Ocular drug	Eye	Thiolated PAsp	[149]
	Glutathione or oxytocin	Colon	Succinic derivative of insulin cross linked pAspHz	[152]
	DTX	Drug resistant cancer cell	PEG-*b*-PAsp	[162]
Glu	Dox	Intracellular tumor	mPEG-*b*-p(LGA-*co*-CELG)	[182]
	Adriamycin hydrochloride	Tumor pH	Star-block-copolymer of PEI, PLGA and PEG	[183]
	Dox	Lysosome	PLGA-*co*-PLA-DPPE	[187]

^a^ See abbreviations for definitions of the terms used.

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
