# Peer review of "pH-Responsive Polypeptide-Based Smart Nano-Carriers for Theranostic Applications"

_molecules, 2019, doi:10.3390/molecules24162961_

Round 1

Reviewer 1 Report

This work provides a very interesting and extensive review of recent studies in the field of nano-carriers with pH-responsive polypeptides for medical applications. In my opinion, the authors presented a detailed description of practically all important results in the area. As a shortcoming of the study, I would mention too short overlook in the conclusion section. I would recommend to add several paragraphs focusing on unsolved problems and future perspectives in this important area of research.

Author Response

We appreciate the reviewer for the valuable suggestion. As the reviewer suggested, we have revised conclusion parts by adding few more points regarding the unsolved problems and future perspectives of pH-responsive polypeptides (refer to p. 29). Corrections are highlighted using the track changes option.

Reviewer 2 Report

Augustine et al. present a review of pH-responsive polypeptide-based nanostructures for biomedical application. This manuscript summarizes recent advances in synthesis of responsive polymeric materials as carriers. Furthermore, the authors describe the application of these carriers in organ, tissue and cell level. Finally, the authors discuss the current challenges and potential future directions in this research field. Overall, this is an interesting and well-written review. A pleasure to read. It is valuable for those looking for an introduction to polypeptide-based drug delivery systems and would be a great resource for researchers as they explore or enter this area. To make this article more stand out, my only suggestion is to provide necessary backgrounds and basic concepts of amino acids, peptides and secondary structures, especially for non-experts. Other than that, I would like to support its publication in Molecules.

Author Response

According to the reviewer’s valuable suggestion, we have included the basic concepts of amino acids, peptides and secondary structures in the introduction section (p. 2). The corrections are highlighted using the track changes options.

Reviewer 3 Report

The authors present an interesting review article on pH-responsive Polypeptide-Based Smart Nano carriers for Theranostic Application. The review is well structured and written and should be of interest to a broad readership. The figures and tables are useful to the reader. After a few edits I think this will be well cited.

The abstract is clear and concise.
The body text is clear and well structured.
The conclusion is clear and concise.
The references are balanced.

Table 1: please expand to be exhaustive for the systems described in the review. This might require the table to be broken down into more than 1 table.

Author Response

According to the reviewer’s valuable suggestion, we found that only Table 1 is not enough to show our concept. For the readers’ clear understanding, we have added new tables, Table 1 and Table 3. In addition, Table 1 is divided to two tables, Table 2 and Table 4. The new tables and the edited tables are highlighted using track changes option in pages 12, 14, 23 and 24.